# Southern Ocean bottom water cooling and ice sheet expansion during the middle Miocene climate transition

Thomas J. Leutert[1,2*], Sevasti Modestou[1,2], Stefano M. Bernasconi[3], A. Nele Meckler[1,2]

[1]Bjerknes Centre for Climate Research, Bergen, 5007, Norway
[2]Department of Earth Science, University of Bergen, Bergen, 5007, Norway
[3]Geological Institute, ETH Zurich, Zurich, 8092, Switzerland

[*]Present address: Max Planck Institute for Chemistry, Mainz, 55128, Germany

*Correspondence to*: Thomas J. Leutert (Thomas.Leutert@mpic.de)

**Abstract.** The middle Miocene climate transition (MMCT), around 14 million years ago (Ma), was associated with a significant climatic shift, but the mechanisms triggering the event remain enigmatic. We present a clumped isotope ($\Delta_{47}$) bottom water temperature (BWT) record from 16.0 Ma to 12.2 Ma from Ocean Drilling Program (ODP) Site 747 in the Southern Ocean, and compare it to existing BWT records from different latitudes. We show that BWTs in the Southern Ocean reached 8–10°C during the Miocene climatic optimum. These high BWT values indicate considerably warmer bottom water conditions than today. Nonetheless, bottom water $\delta^{18}O$ (calculated from foraminiferal $\delta^{18}O$ and $\Delta_{47}$) suggests substantial amounts of land ice throughout the interval of the study. Our dataset further demonstrates that BWTs at Site 747 were variable with an overall cooling trend across the MMCT. Notably, a cooling of around 3–5°C preceded the stepped main increase in benthic $\delta^{18}O$, interpreted as global ice volume expansion, and appears to have been followed by a transient bottom water warming starting during or slightly after the main ice volume increase. We speculate that a regional freshening of the upper water column at this time may have increased stratification and reduced bottom water heat loss to the atmosphere, countering global cooling in the bottom waters of the Southern Ocean and possibly even at larger scales. Feedbacks required for substantial ice growth and/or tectonic processes may have contributed to the observed decoupling of global ice volume and Southern Ocean BWT.

## 1 Introduction

During the Cenozoic Era (the last 65 Myr), Earth's climate transitioned from a state of expansive warmth with very limited ice to colder conditions and permanent ice sheets at the poles (Zachos et al., 2001). The middle Miocene climate transition (MMCT, ~14.5–13 Ma) represents one of the main steps of Cenozoic climate reorganization (e.g., Flower and Kennett, 1993; Super et al., 2018). A substantial increase in benthic foraminiferal oxygen isotope ratios ($\delta^{18}O$) during the MMCT has

been interpreted to reflect a combination of decreasing bottom water temperatures (BWTs) and ice sheet expansion (increasing bottom water $\delta^{18}O$) occurring in the Southern Hemisphere (Lear et al., 2015; Lewis et al., 2007). A roughly coeval decrease in atmospheric $pCO_2$ of ~100–300 ppm was estimated based on boron isotope and alkenone records, suggesting a coupling of $pCO_2$ and benthic foraminiferal $\delta^{18}O$ during this interval (Foster et al., 2012; Sosdian et al., 2018; Super et al., 2018). Atmospheric $pCO_2$ also appears to be coupled to upper ocean temperatures in the North Atlantic and Southern Ocean (Leutert et al., 2020; Super et al., 2018). Conversely, several studies propose a degree of decoupling between BWT and global ice volume during the middle Miocene (Billups and Schrag, 2002; Lear et al., 2010, 2015; Shevenell et al., 2008). These studies are based on deconvolving the bottom water $\delta^{18}O$ ($\delta^{18}O_{bw}$) and temperature signals in benthic foraminiferal $\delta^{18}O$ with independent temperature estimates based on benthic foraminiferal Mg/Ca ratios. Their results indicate a middle Miocene decrease in BWT of ~0.5–3°C. Taking into account the $\delta^{18}O$ increase of roughly 1 ‰ in benthic foraminifera, this cooling would imply a drop in global sea level of ~30–110 m, based on the Pleistocene seawater $\delta^{18}O$-sea level calibration of 0.08–0.11 ‰ per 10 m sea level (Fairbanks and Matthews, 1978; Lear et al., 2010) and the oxygen isotope temperature equation (Eq. (9)) of Marchitto et al. (2014). More advanced approaches using backstripping and different modelling techniques suggest a sea level drop of ~20–40 m across the MMCT (de Boer et al., 2010; Frigola et al., 2018; Gasson et al., 2016; Kominz et al., 2008; Langebroek et al., 2009).

Although the MMCT represents one of the most fundamental reorganizations in global climate during the Cenozoic era (e.g., Flower and Kennett, 1993; Zachos et al., 2001), there are still major uncertainties associated with estimating the magnitude and timing of BWT and global ice volume changes. These uncertainties are mainly caused by the small number of independent BWT records resulting in limited spatial and temporal coverage for the middle Miocene, but also by current limitations of the applied temperature proxies. Middle Miocene data coverage is especially poor in the high-latitude Southern Ocean, where high-resolution BWT records are conspicuously lacking. An existing lower-resolution (~200–300 kyr) Southern Ocean proxy record based on Mg/Ca signatures of benthic foraminiferal tests from Ocean Drilling Program (ODP) Site 747 indicates a bottom water cooling of ~2–3°C from around 15 Ma to 12 Ma (Billups and Schrag, 2002). However, the middle Miocene portion of this BWT record from ODP Site 747 does not have the temporal resolution to adequately capture the magnitude and timing of BWT changes across the MMCT. Furthermore, the application of the Mg/Ca thermometer to middle Miocene benthic foraminifera is complicated by a number of non-thermal effects. Notable amongst these are differential vital effects in foraminifera (e.g., Lear et al., 2002) and the effect of seawater Mg/Ca that has not remained constant on timescales longer than several million years (Evans and Müller, 2012). Finally, benthic foraminiferal Mg/Ca signatures can be influenced by changes in carbonate ion saturation state, especially at low saturation (Elderfield et al., 2006; Lear et al., 2010; Yu and Elderfield, 2008). Previous studies have attempted to minimize saturation state-related effects on Mg/Ca by using only infaunal foraminifera (e.g., *Oridorsalis umbonatus*) precipitating their tests in pore waters that may be buffered to some extent against carbonate saturation changes (Elderfield et al., 2006; Lear et al., 2015) and/or by correcting for changes in saturation state based on paired Mg/Ca and Li/Ca measurements (Lear et al., 2010). Nevertheless, the impact

of fluctuating saturation states on middle Miocene Mg/Ca signatures remains controversial. Independent temperature records are required to better understand the mechanisms controlling the Southern Ocean climate evolution during this interval of global change.

The carbonate clumped isotope ($\Delta_{47}$) paleothermometer is based on the measured abundance of $^{13}C$–$^{18}O$ bonds relative to their stochastic distribution (Ghosh et al., 2006; Schauble et al., 2006), and is independent of the isotopic composition of the parent water from which the carbonate grew (e.g., Eiler, 2011). On the basis of current knowledge, other environmental variables such as pH and salinity appear to be of minor importance for measured $\Delta_{47}$ values over the range of natural variation (Tripati et al., 2015; Watkins and Hunt, 2015). When applied to foraminiferal calcite, the method also does not show detectable species-specific vital effects (Grauel et al., 2013; Meinicke et al., 2020; Modestou et al., 2020; Peral et al., 2018; Piasecki et al., 2019; Tripati et al., 2010). Diagenetic effects on benthic foraminiferal $\Delta_{47}$ signatures cannot be excluded in all depositional environments, similar to the benthic foraminifer-based Mg/Ca thermometer. However, a first study targeting the impact of diagenetic alteration on $\Delta_{47}$ in foraminiferal tests indicated a low sensitivity of benthic foraminiferal $\Delta_{47}$ values to diagenesis in settings commonly used for Cenozoic climate reconstructions (Leutert et al., 2019). Consequently, the $\Delta_{47}$ thermometer holds great promise for reconstructing accurate BWTs from benthic foraminiferal tests, despite comparably large analytical uncertainties and sample mass requirements (e.g., Leutert et al., 2019). The $\Delta_{47}$ thermometer has been previously applied to middle Miocene benthic foraminifera from ODP Site 761 in the Indian Ocean yielding results that are in good agreement with Mg/Ca BWTs from the same site (Lear et al., 2010; Modestou et al., 2020). However, there are intervals with very low temporal resolution and potential hiatuses in the middle Miocene record from Site 761, limiting its informative value for understanding the drivers of the MMCT. Here, we present $\Delta_{47}$-based BWTs measured on benthic foraminiferal calcite from ODP Site 747 located on the Kerguelen Plateau in the Indian Ocean sector of the Southern Ocean (Fig. 1). While sediment samples were taken at a relatively high temporal resolution (~20 kyr), temperature information is provided at lower resolution but with minimal aliasing errors. We compare our new absolute BWT record to previous BWT estimates for the middle Miocene, and interpret the BWT records in the context of middle Miocene glaciation and $CO_2$ drawdown.

## 2 Material and methods

### 2.1 Site details

ODP Site 747 (54°48.68'S, 76°47.64'E; 1695 m water depth) lies on the Kerguelen Plateau in the Southern Ocean (Fig. 1; Schlich et al., 1989). At present, the site is situated south of the Polar Front and is bathed by Circumpolar Deep Water (CDW) with a temperature of ~1–2°C (Belkin and Gordon, 1996; Billups and Schrag, 2002). The middle Miocene geographic position of Site 747 relative to Antarctica was similar to today (e.g., Abrajevitch et al., 2014) with a paleolatitude

between 51°S and 56°S at 16–12 Ma (van Hinsbergen et al., 2015; Torsvik et al., 2012). Middle Miocene benthic foraminiferal species found at Site 747 are indicative of a lower bathyal to abyssal depth at that time (Schlich et al., 1989). The clumped isotope record generated in this study covers the depth interval from 62.64 m below sea floor (mbsf, Sample 747A-7H-5, 14–16 cm) to 85.36 mbsf (Sample 747A-9H-8, 75–77 cm) in Hole 747A. 191 samples (15–20 cm$^3$, mostly calcareous nannofossil ooze with foraminifera) were taken continuously with a mean temporal resolution of

around 20 kyr (Table S1). We slightly rescaled the originally assigned shipboard sample depths to account for core expansion (Table 1 of Schlich et al. (1989)), similar to previous studies focusing on the middle Miocene section of Hole 747A (e.g., Abrajevitch et al., 2014; Majewski and Bohaty, 2010).

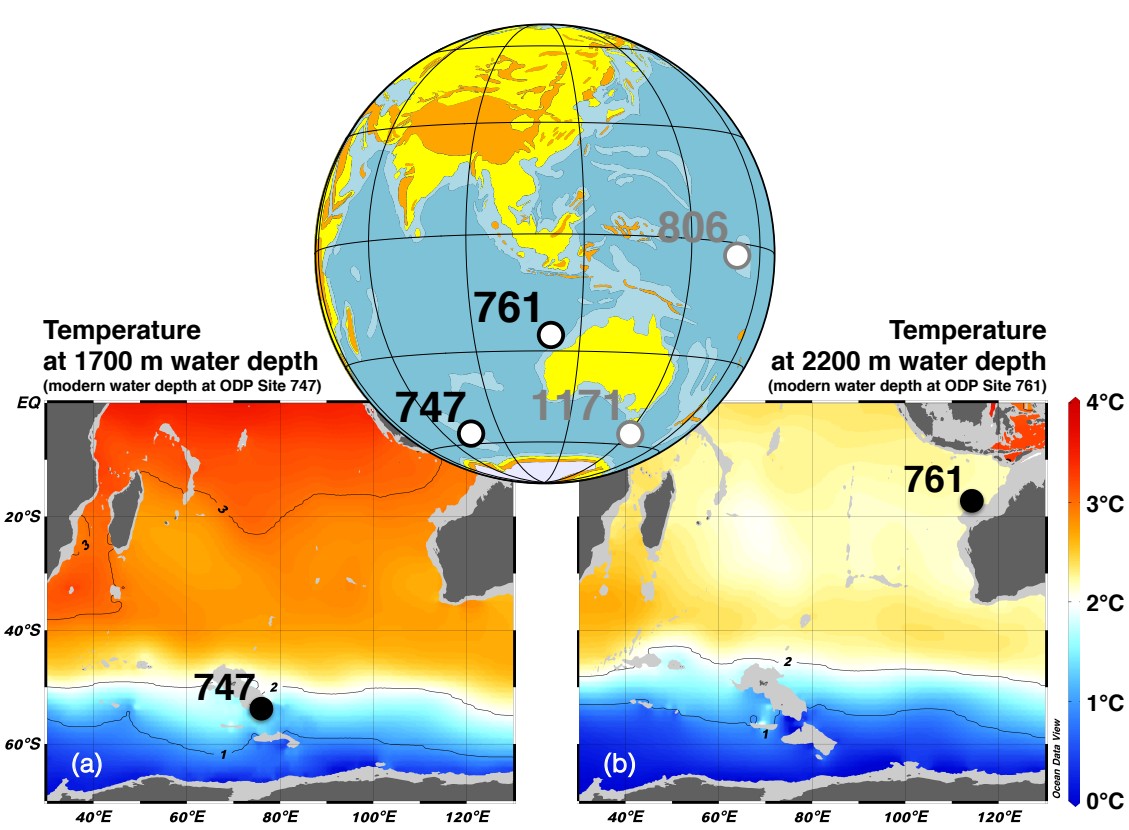

**Fig. 1:** Ocean temperatures at modern water depths and paleogeographic reconstruction for 14 Ma. Modern water depths of ODP Sites 747 and 761 are ~1700 m and ~2200 m, respectively (Lear et al., 2010; Schlich et al., 1989). Maps of annual mean temperatures at these depths are shown in (**a**) and (**b**). Temperatures from the 2013 World Ocean Atlas (Locarnini et al., 2013) visualized with Ocean Data View (Schlitzer, 2019). Inset map with paleogeographic reconstruction (deep ocean: dark turquoise, shallow marine: light turquoise, landmass: yellow, mountain: orange, ice sheet: light purple) created with GPlates (Cao et al., 2017; Matthews et al., 2016; Müller et al., 2018).

## 2.2 Age models

We revised the Hole 747A age model by integrating six magnetostratigraphic tie points (Abrajevitch et al., 2014; Majewski and Bohaty, 2010) on the GTS2012 timescale (Gradstein et al., 2012), three benthic foraminiferal $\delta^{13}$C-based tie points associated with the "Monterey" carbon-isotope excursion (using the nomenclature of Holbourn et al. (2007)), and one peak warm event visible in benthic foraminiferal $\delta^{13}$C and $\delta^{18}$O (Kochhann et al., 2016) (Fig. S1 and Table S2). For the $\delta^{13}$C-based tie points, we used the high-resolution isotope stratigraphies of IODP Sites U1335, U1337 and U1338 in the eastern equatorial Pacific Ocean (Holbourn et al., 2014; Kochhann et al., 2016; Tian et al., 2018) as reference (Fig. 2c). In addition, we included a hiatus at the core break between Cores 7H and 8H, identified by previous studies (e.g., Majewski and Bohaty, 2010). $\delta^{18}$O and $\delta^{13}$C time series of Sites 747, 761, 806, U1335, U1337 and U1338 are shown in Fig. 2 with isotope-based age tie points for Site 747 as black crosses. The age models for ODP Sites 761 and 1171 (not shown) are from Leutert et al. (2020). For ODP Site 806, we utilized a previously published orbitally tuned age model from ~14.1 Ma to ~13.3 Ma. For the older and younger parts of the Site 806 record (~16.6–14.1 Ma and ~13.3–11.6 Ma), we updated biostratigraphic events from Kroenke et al. (1991) and Chaisson and Leckie (1993) to the GTS2012 timescale (Gradstein et al., 2012), and applied polynomial curve fits (Fig. S2 and Table S3).

## 2.3 Sample material

Each sediment sample was freeze-dried, washed over a 63 µm sieve, oven-dried at 50°C and then dry-sieved into different size fractions. We mainly picked tests of *Cibicidoides mundulus* from the 250–355 µm size fraction for our measurements. For samples with low abundances of benthic foraminifera in this size fraction, the >355 µm size fraction was also included. The interval from ~16.0 Ma to ~15.3 Ma was additionally complemented with measurements on *Cibicidoides wuellerstorfi*. No inter-species offsets in benthic foraminiferal $\Delta_{47}$ have been found in previous studies (e.g., Modestou et al., 2020; Piasecki et al., 2019). To assess inter-species $\delta^{18}$O and $\delta^{13}$C offsets, however, both *Cibicidoides* species were measured separately in 36 sediment samples (Table S1). Middle Miocene benthic foraminifera (and more specifically *Cibicidoides*) from Site 747 were previously described as well preserved (e.g., Abrajevitch et al., 2014; Billups and Schrag, 2002) and our examination confirms this impression (Figs. S3 and S4). We note that some of the analysed specimens of *C. mundulus* and *C. wuellerstorfi* closely resemble the *sensu lato* morphotype of the respective species (shown in Fig. 2 of Gottschalk et al., 2016).

Prior to isotope analysis, we cracked open the picked specimens and ultrasonicated the test fragments in deionized water (3×30 seconds) and methanol (1×10–30 seconds) to remove adhering sediment. Test fragments were rinsed with deionized water once between each ultrasonication step and at least three times at the end of the cleaning. The cleaned test fragments were subsequently oven-dried at 50°C.

## 2.4 Isotope measurements and data processing

Low abundances of carbonate ions containing both [13]C and [18]O isotopes require stringent analytical procedures and comparably large sample sizes to obtain clumped isotope temperatures that are precise enough for Cenozoic ocean temperature reconstructions. We achieve the necessary precision by averaging over ~30–40 clumped isotope values measured on small (~100 μg) carbonate samples (Fernandez et al., 2017; Hu et al., 2014; Meckler et al., 2014; Schmid and Bernasconi, 2010). Results from adjacent samples are pooled to achieve this number of measurements (e.g., Grauel et al., 2013; Rodríguez-Sanz et al., 2017), due to the generally low abundance of mono-specific benthic foraminifera (allowing for only 1–5 individual measurements per sample, Fig. S5b). Producing a low-resolution clumped isotope temperature record with this approach yields higher-resolution $\delta^{18}O$ and $\delta^{13}C$ time series in parallel (Tables S1 and S4).

Clumped isotope measurements were performed using two Thermo Scientific MAT 253 Plus mass spectrometers at the University of Bergen, Norway, and one Thermo Scientific MAT 253 mass spectrometer at ETH Zurich, Switzerland. All mass spectrometers were coupled to Thermo Fisher Scientific Kiel IV carbonate preparation devices. $CO_2$ gas was extracted from carbonate samples with phosphoric acid at a reaction temperature of 70°C. A Porapak trap included in each Kiel IV carbonate preparation system was kept at –20°C to remove organic contaminants from the sample gas (Schmid et al., 2012). Between each run, the Porapak trap was heated at 120°C for at least one hour for cleaning. Every measurement run included a similar number of samples and carbonate standards. Four carbonate standards (ETH-1, ETH-2, ETH-3 and ETH-4) with different isotopic compositions and ordering states were used for monitoring and correction of the results (Table S5). External reproducibilities (one standard deviation) in corrected $\Delta_{47}$ values of ETH-1, ETH-2, ETH-3 and ETH-4 were typically between 0.030 ‰ and 0.040 ‰ (Table S6). External reproducibilities (one standard deviation) for $\delta^{18}O$ and $\delta^{13}C$ values of the same standards (given relative to VPDB) were 0.03–0.10 ‰ and 0.02–0.06 ‰, respectively. More details on isotope analysis and data processing can be found in Appendix A.

We converted the sample $\Delta_{47}$ values (averages over ~30–40 separate measurements each) into temperature (T, in °C) using a calibration based on various recent datasets from core top-derived foraminifera, corrected with the same carbonate standards as used in our study (Eq. (2) of Meinicke et al. (2020)):

$$T = \sqrt{\frac{0.0431 \times 10^6}{\Delta_{47} - 0.1876}} - 273.15 \qquad (1)$$

This combined calibration has been recommended for foraminifer samples (Meinicke et al., 2020). We note that the individual datasets in this compilation (Meinicke et al., 2020; Peral et al., 2018; Piasecki et al., 2019) are all in good agreement with a travertine-based calibration (Kele et al. (2015), recalculated by Bernasconi et al. (2018)) spanning a wider

temperature range (6–95°C). For consistency, previously published $\Delta_{47}$-based ocean temperatures from ODP Sites 761 (Modestou et al., 2020) and 1171 (Leutert et al., 2020) originally based on the travertine calibration were recalculated with the calibration equation of Meinicke et al. (2020) (Tables S7 and S8).

    The $\Delta_{47}$ signal from individual analyses (Fig. S5a) is by nature much noisier in comparison to $\delta^{18}O$ and $\delta^{13}C$ (Fig. 2), necessitating an averaging of $\Delta_{47}$ over many adjacent samples before interpreting the data in terms of calcification
temperature. We have averaged our $\Delta_{47}$ data using two approaches (Fig. 3), each with different advantages: (1) We averaged results from around 30–40 individual measurements from neighbouring samples, avoiding averaging across hiatuses and intervals with no measurements. These BWT averages are shown as filled circles, with horizontal lines indicating the averaging intervals (circles are plotted at average ages of the respective groups of measurements) and vertical lines indicating 68 % (solid) and 95 % (dashed) confidence intervals. The number of measurements used for the calculation of
each mean temperature value is listed at the top of Fig. 3. (2) 400 kyr-moving averages based on 30 or more measurements are shown as solid lines, whereas those based on fewer measurements are dotted. The latter approach does not require a decision on each averaging interval, and may thus be better suited for inter-site comparison. We note that small-scale features in the moving average curves (around 1°C or less) are likely caused by the scatter in the underlying individual $\Delta_{47}$ measurements, and should not be interpreted as real climate signals. Furthermore, signal changes during rapid transitions can
be "smoothed out" to some extent. A comparison of our smoothed clumped isotope temperature curves to different LOESS non-parametric regressions of the data is shown in Fig. S6. We propagated analytical and calibration uncertainties in $\Delta_{47}$-based temperatures (as described in Appendix A of this study and the supporting information of Huntington et al. (2009)), and report combined uncertainties as 68 % and 95 % confidence intervals.

    $\Delta_{47}$-based BWTs were used in combination with benthic foraminiferal $\delta^{18}O$ ($\delta^{18}O_{foram}$) to calculate $\delta^{18}O_{bw}$ (reported relative
to VSMOW) with Eq. (9) of Marchitto et al. (2014):

$$\delta^{18}O_{bw} = \delta^{18}O_{foram} + 0.27 + 0.245 \times BWT - 0.0011 \times BWT^2 - 3.58 \qquad (2)$$

    For these calculations, benthic foraminiferal $\delta^{18}O$ values of the taxon *Cibicidoides* were averaged over the same intervals as
have been used for $\Delta_{47}$ averaging. For Site 747, we used the $\delta^{18}O$ values from this study (measured on *C. mundulus* and *C. wuellerstorfi*), whereas the foraminiferal $\delta^{18}O$ values for Site 761 were compiled from existing studies (Holbourn et al., 2004; Lear et al., 2010; Modestou et al., 2020). Alternative oxygen isotope temperature equations were also tested (Fig. S7).

# 3 Results

## 3.1 Benthic foraminiferal $\delta^{18}O$ and $\delta^{13}C$ values

The isotope records of Site 747 (Fig. 2) displays features typical of middle Miocene sequences, including the stepped increase in benthic $\delta^{18}O$ across the MMCT and the pronounced $\delta^{13}C$ maxima associated with the "Monterey" carbon isotope excursion (e.g., Holbourn et al., 2007, 2014; Kochhann et al., 2016; Vincent and Berger, 1985). From ~16.0 Ma to ~15.3 Ma, we analysed stable isotope compositions of both *C. mundulus* and *C. wuellerstorfi*, allowing for a direct assessment of species-specific effects on the isotopic compositions of these two different epifaunal species (Fig. 2a and c). $\delta^{18}O$ values

measured on *C. mundulus* and *C. wuellerstorfi* appear indistinguishable, whereas a consistent offset of up to ~0.5 ‰ exists between the $\delta^{13}C$ values of these species at Site 747. Similar $\delta^{13}C$ offsets between *C. mundulus* and *C. wuellerstorfi* have been previously observed for the sub-Antarctic Atlantic during the Quaternary (Gottschalk et al., 2016). Our $\delta^{13}C$ values from the middle Miocene underscore the need to carefully examine inter-species offsets in $\delta^{13}C$ before combining different species to produce a single $\delta^{13}C$ curve.

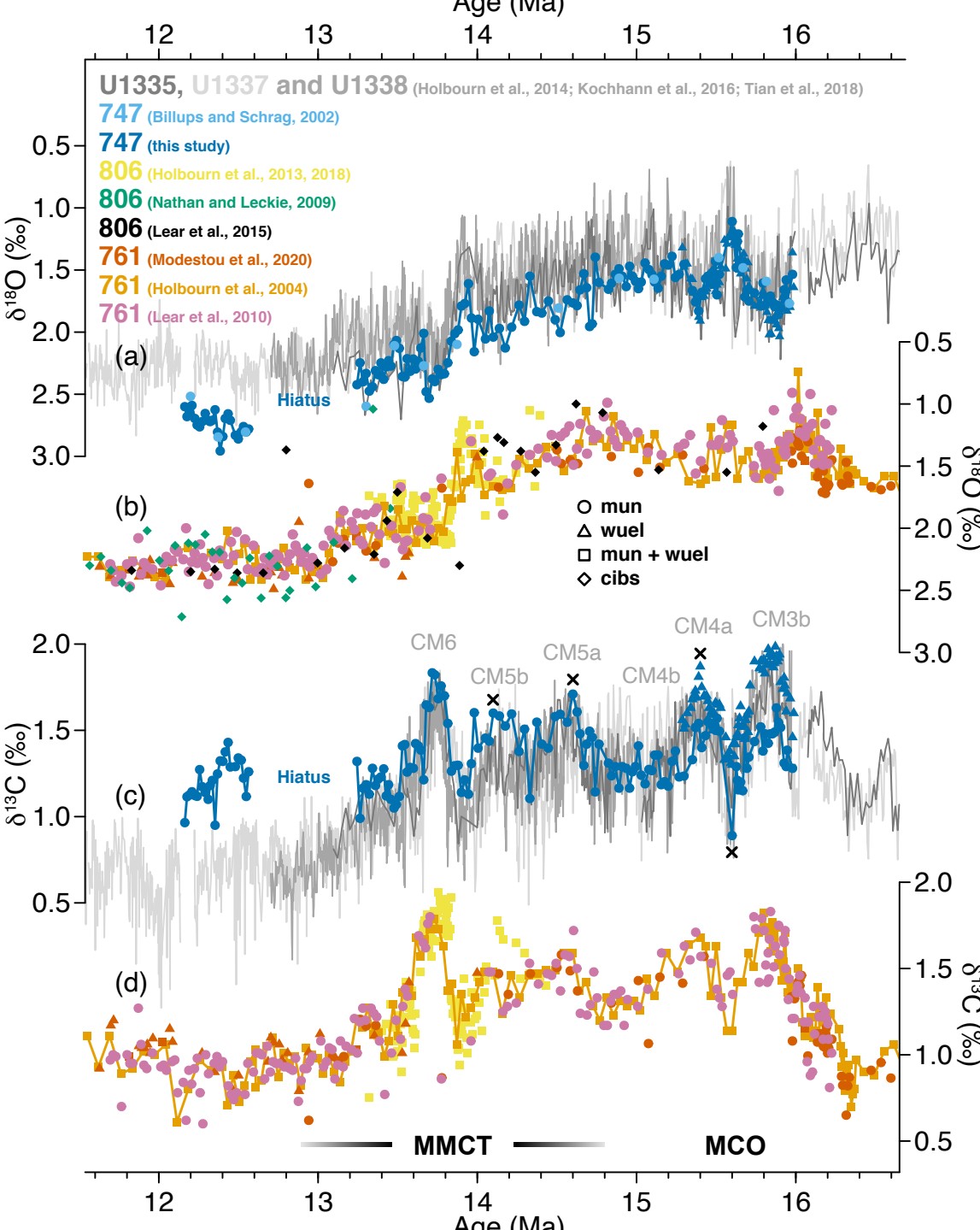

**Fig. 2:** Comparison of benthic isotope data. Benthic foraminiferal $\delta^{18}O$ (**a, b**) and $\delta^{13}C$ (**c, d**) records are shown from ODP Site 747 in the Southern Ocean (Billups and Schrag, 2002; this study), ODP Site 761 in the eastern Indian Ocean (Holbourn et al., 2004; Lear et al., 2010; Modestou et al., 2020), ODP Site 806 in the western equatorial Pacific (Holbourn et al., 2013, 2018; Lear et al., 2015; Nathan and Leckie, 2009) as well as IODP Sites U1335, U1337 and U1338 in the eastern equatorial Pacific Ocean (Holbourn et al., 2014; Kochhann et al., 2016; Tian et al., 2018). Correlation tie points for Site 747 (this study) are visualized with black crosses. We only plot $\delta^{18}O$ and $\delta^{13}C$ values from Sites 747 and 761 that were measured on the species *C. mundulus* (mun) and *C. wuellerstorfi* (wuel). In contrast to Site 747, offsets in both $\delta^{18}O$ and $\delta^{13}C$ between these species appear minimal at Site 761 (Holbourn et al., 2004). We note that we also use $\Delta_{47}$ values from other benthic foraminiferal species from Site 761 (see Modestou et al. (2020) for details), as no species-specific vital effects on benthic foraminiferal $\Delta_{47}$ have been observed (Modestou et al., 2020; Piasecki et al., 2019). For Site 806, we show $\delta^{18}O$ values of *Cibicidoides* spp. (cibs) (Lear et al., 2015; Nathan and Leckie, 2009), in addition to $\delta^{18}O$ and $\delta^{13}C$ measured specifically on tests of *C. mundulus* and *C. wuellerstorfi* (Holbourn et al., 2013, 2018). $\delta^{18}O$, $\delta^{13}C$ and $\Delta_{47}$ at Sites 747 and 761 were measured several times per sample in this study and Modestou et al. (2020). See Fig. S5 for $\Delta_{47}$ values and number of replicate measurements for each sediment sample.

## 3.2 Clumped isotope bottom water temperatures at Site 747

Independent of the averaging approach, $\Delta_{47}$-based BWTs at Site 747 are highest (8.9 ± 1.3°C, uncertainties 95% confidence level) from around 16.0 Ma to 14.4 Ma during the Miocene climatic optimum (MCO; Fig. 3b). Thereafter, during the early phase of the MMCT, BWTs decrease by 4.2 ± 2.3°C (difference between mean BWT value from ~16.0 Ma to ~14.4 Ma and mean BWT value from ~14.4 Ma to ~13.6 Ma). The cooling appears to partly coincide with an overall increase in benthic foraminiferal $\delta^{18}O$ from around 15 Ma to 14 Ma, reflecting bottom water cooling and/or global ice sheet growth. However, the $\Delta_{47}$-based cooling is much more pronounced than the $\delta^{18}O$ data would suggest. Even if the $\delta^{18}O$ signal was influenced by BWT only, then the gradual ~0.5 ‰ increase in benthic $\delta^{18}O$ would correspond to a cooling of roughly 2°C (e.g., Marchitto et al., 2014). During the subsequent distinct stepped increase in benthic $\delta^{18}O$ around 13.9–13.7 Ma, the $\Delta_{47}$-based BWT record on the other hand does not provide any evidence for a significant cooling. To the contrary, the Site 747 $\Delta_{47}$ record reveals a transient warming starting at or just after the stepped benthic $\delta^{18}O$ increase. The magnitude of this transient warming in the later phase of the MMCT is 3.2 ± 3.1°C (difference between mean BWT estimates for ~14.0–13.6 Ma and ~13.6–13.2 Ma). The warming appears to some extent also imprinted in the benthic $\delta^{18}O$ signal at Site 747, as visible in the slight $\delta^{18}O$ decrease between 13.7 Ma and 13.6 Ma. A hiatus prevents us from drawing any inferences about bottom water conditions from ~13.2 Ma to ~12.6 Ma. In the youngest interval covered by our study (~12.6–12.2 Ma), bottom water conditions are comparably cold again (5.8 ± 2.1°C).

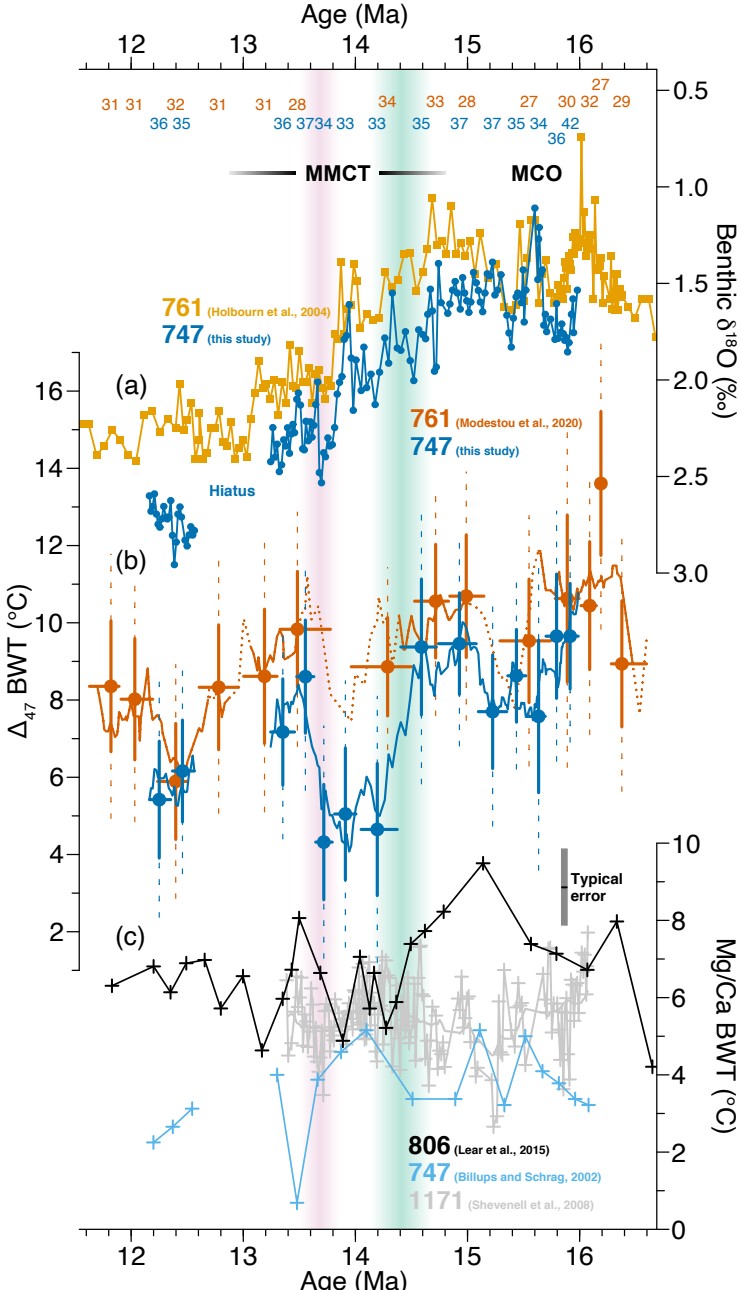

**Fig. 3:** Comparison of benthic foraminiferal $\delta^{18}O$ and $\Delta_{47}$-based bottom water temperatures (BWTs) from ODP Sites 747 and 761 with Mg/Ca-derived BWTs from ODP Sites 747, 806 and 1171. **(a)** Benthic foraminiferal $\delta^{18}O$ from Sites 747 and 761. **(b)** $\Delta_{47}$-based BWTs based on averages of >30 $\Delta_{47}$ measurements each are shown as filled circles (horizontal solid lines: averaging intervals, vertical solid lines: 68 % confidence intervals, vertical dashed lines: 95 % confidence intervals). The marked BWT decrease during the early phase of the MMCT and the transient bottom water warming during the later phase of the MMCT are marked with light green and purple vertical bars,

respectively. The number of measurements used for each average is shown at the top of the plot. The position on the x-axis shows the average age of each temperature value. 400 kyr-moving averages based on at least 30 and fewer than 30 measurements are shown as solid and dotted lines, respectively. Note that rapid fluctuations (of around 1°C) in these moving averages should not be interpreted in terms of climate (see Discussion). (**c**) Mg/Ca temperatures from Sites 747 and 1171 are as published previously (Billups and Schrag, 2002; Shevenell et al., 2008). For Site 806, temperatures were calculated from infaunal foraminiferal Mg/Ca (Lear et al., 2015) using seawater Mg/Ca (polynomial curve fit through compiled seawater Mg/Ca records) and the linear temperature calibration of Lear et al. (2015). In addition, we illustrate the typical uncertainty introduced by sample reproducibility and calibration errors (±1°C, vertical black bar) (Lear et al., 2015).

## 4 Discussion

### 4.1 Comparison between different bottom water temperature estimates

Comparison of our $\Delta_{47}$-based BWTs from Site 747 with $\Delta_{47}$-based BWTs from Site 761 off northwest Australia in the Indian Ocean (Modestou et al., 2020) reveals good agreement, where temperatures are based on at least 30 $\Delta_{47}$ measurements (solid lines of the moving averages), with the Site 747 BWTs being slightly lower. Temperature averages from <30 measurements (dotted lines) are less certain, and thus not focus of our interpretation here (see Material and Methods). Note that we processed the $\Delta_{47}$ measurement values from Site 761 (Modestou et al., 2020) in the same way as our results from Site 747 (e.g., temperature calibration, smoothing) to optimize comparability of BWTs from these two middle Miocene reference sites. Since modern BWTs at Sites 747 and 761 are similar (~1–3°C; see Fig. 1), we expect middle Miocene temperature differences between Sites 747 and 761 to also be small, although the middle Miocene water depths of these sites may have been somewhat different from today. Our study confirms the similarity of BWTs at these sites for large parts of the studied interval, suggesting a close to modern meridional temperature gradient around 2000 m water depth in a scenario of substantially (by up to ~9°C) warmer bottom waters. Unfortunately, the period of most pronounced BWT change at Site 747 during the MMCT is characterized by very low data density at Site 761, due to low benthic foraminiferal abundances resulting in few measurements, and possibly a hiatus (core break between Cores 5H and 6H from Site 761 around 14.1 Ma). This leaves open the question whether the substantial early MMCT cooling around 14.5–14.0 Ma and the subsequent warming were restricted to particular regions in the Southern Ocean, or whether they were more widespread features.

Interestingly, a Mg/Ca record of the infaunal benthic foraminifer *O. umbonatus* from ODP Site 806 in the equatorial Pacific (present water depth 2521 m; Lear et al., 2015) indicates BWT trends that are similar to those reconstructed from $\Delta_{47}$ at Site 747 during the MMCT (Fig. 3). Even though the Site 806 Mg/Ca record is of limited temporal resolution (~100–200 kyr), this low latitude record provides evidence that the early cooling and the subsequent warming reconstructed at Site 747 could have indeed been of larger scale or even global significance.

Other available Mg/Ca-based BWT records covering the MMCT do not show the same features. Similar to Site 806, Mg/Ca ratios were also measured on the infaunal species *O. umbonatus* at Site 761 (Lear et al., 2010). This approach yields BWTs that are within uncertainty of those from $\Delta_{47}$ measured at the same site (Fig. S8; regardless of whether or not the Mg/Ca-based BWTs have been corrected for changes in saturation state (Modestou et al., 2020)), and show no indication for the substantial BWT changes derived from $\Delta_{47}$ at Site 747 and Mg/Ca at Site 806 (Lear et al., 2015). However, we note that Mg/Ca-based BWT estimates from Site 761 have been deemed less reliable than those from Site 806, due to unusual and variable pore water chemistry at Site 761. Mg/Ca records from Southern Ocean Sites 747 (Kerguelen Plateau; Billups and Schrag, 2002) and 1171 (South Tasman Rise; Shevenell et al., 2008) measured on the epifaunal species *C. mundulus* also do not show the large temperature swings (Fig. 3). The observed discrepancies between the sites could suggest a regional and/or depth-related differentiation in water mass properties, related to transient ocean circulation changes during the MMCT (see further discussion below). However, especially in the light of the discrepancies between BWTs estimated from Mg/Ca (Billups and Schrag, 2002) and $\Delta_{47}$ at Site 747, another possible explanation is additional non-thermal controls on Mg/Ca and/or $\Delta_{47}$, which may be related to seawater chemistry during test precipitation and/or post-depositional alteration, such as dissolution.

To the best of our current knowledge, seawater chemistry does not appear to significantly influence $\Delta_{47}$ signatures in foraminifera over the range of natural variation (e.g., Tripati et al., 2015; Watkins and Hunt, 2015). On the other hand, it has been shown that Mg/Ca signatures can be affected by changes in seawater Mg/Ca (Evans and Müller, 2012) and carbonate ion saturation (Elderfield et al., 2006; Yu and Elderfield, 2008). On the timescales considered here, the latter is more likely to be important. The relatively few Mg/Ca-based BWTs from Site 747 can be directly compared to our BWTs based on $\Delta_{47}$ from the same site (Fig 4a). $\Delta_{47}$- and Mg/Ca-based BWTs appear to diverge most pronouncedly in times of increased dissolution (high percentage of benthic foraminiferal tests and fragments), indicating fluctuations in bottom water carbonate ion saturation (Diester-Haass et al. (2013); Fig. 4b and c). Mg/Ca-based temperatures from Site 747 were measured on foraminiferal tests of the epifaunal species *C. mundulus*; compared to infaunal foraminifera, this species lives in more direct contact with bottom water, and may thus be more prone to saturation state-related effects (Elderfield et al., 2006; Lear et al., 2015). The observation of diverging Mg/Ca- and $\Delta_{47}$-based BWTs in times of increased dissolution supports the interpretation of a possible saturation state effect on the Mg/Ca signatures of *C. mundulus* (see Fig. S9 for sensitivity calculation).

In addition to saturation state effects, variable dissolution itself (Fig. 4b and c) could have influenced foraminiferal Mg/Ca and/or $\Delta_{47}$ signatures. For planktic foraminifera, dissolution controlled by bottom water saturation has the potential to significantly lower initial Mg/Ca signatures and thus also the estimated ocean temperatures in certain burial settings (e.g., Regenberg et al., 2014). Dissolution may also impact the Mg/Ca signatures of benthic foraminiferal tests, although the tests

of benthic foraminifera appear generally denser and more resistant to dissolution than those of planktic foraminifera (e.g., Berger, 1973; Pearson et al., 2001). The effects of dissolution on benthic foraminiferal Mg/Ca have thus received little attention. Similarly, dissolution effects on benthic foraminiferal $\Delta_{47}$ signatures have not yet been specifically assessed. While there is currently no evidence for a significant dissolution effect on foraminiferal $\Delta_{47}$ (e.g., Breitenbach et al., 2018; Leutert et al., 2019) or variable dissolution of benthic foraminiferal calcite at Site 747 during the interval of this study (Fig. S4), a potential effect of dissolution cannot be fully ruled out. We thus note that this aspect warrants further study, but interpret $\Delta_{47}$-based temperatures as unaffected by dissolution in the absence of indications otherwise. The good agreement of our $\Delta_{47}$-based BWT estimates from Site 747 with the infaunal Mg/Ca BWT record from Site 806 (Lear et al., 2015) lends support to this interpretation.

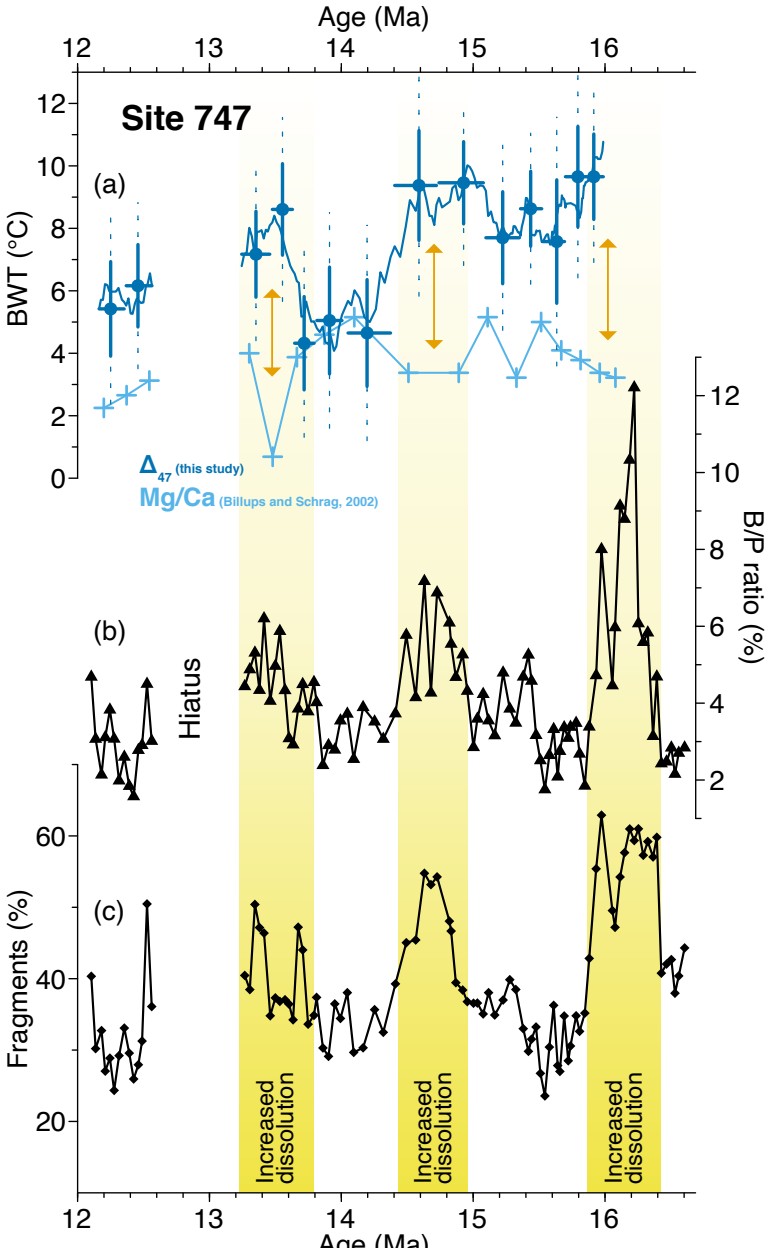

**Fig. 4:** Bottom water temperature (BWT) and dissolution at Site 747. (**a**) $\Delta_{47}$- and Mg/Ca-based BWT estimates (this study; Billups and Schrag, 2002) are shown versus (**b**) percent benthic to planktic (B/P) foraminiferal test ratios (Diester-Haass et al., 2013) and (**c**) percent fragments in a sample (Diester-Haass et al., 2013). Percent fragments and B/P foraminiferal test ratios have been previously used to monitor dissolution at Site 747 (Diester-Haass et al., 2013). Intervals interpreted as affected by increased dissolution of planktic foraminifera are highlighted with yellow bars. Orange arrows indicate intervals where $\Delta_{47}$- and Mg/Ca-based temperature estimates appear to diverge most.

## 4.2 Regional and global implications

The observation of a pronounced early MMCT bottom water cooling and subsequent warming during the later MMCT at Site 747 is surprising, and suggests previously unrecognized changes in deep water properties surrounding one of the major climate transitions in the Cenozoic era. Upper ocean temperature records from the Southern Ocean are sparse, but existing data (Kuhnert et al., 2009; Leutert et al., 2020; Shevenell et al., 2004) do not show similarity to the temperature pattern we reconstruct for the deep ocean. Instead, the multiproxy temperature dataset from ODP Site 1171 on the South Tasman Rise indicates that the cooling in the upper waters of the Southern Ocean was synchronous to the benthic $\delta^{18}O$ increase reflecting a substantial expansion of the Antarctic ice sheet (Fig. 5; Leutert et al., 2020). This observation suggests that the Southern Ocean BWT signal reconstructed from Site 747 benthic foraminiferal $\Delta_{47}$ reflects changes in deep water properties, rather than a high-latitude surface ocean response. Occurring in an interval of overall decreasing $p$CO$_2$ (Foster et al., 2012; Sosdian et al., 2018; Super et al., 2018), the early MMCT deep ocean cooling might reflect ice sheet-related changes in deep ocean circulation. Recent model results suggest that especially the spatial extent of the Antarctic ice sheet may have played an important role for BWT during the MMCT, because albedo changes affect the hydrological cycle and the regions of deep water formation around Antarctica (Bradshaw et al., 2021). Alternatively, circulation changes at that time could have been related to tectonic processes accompanying the opening of the Drake Passage and Scotia Sea (e.g., Dalziel et al., 2013; Lagabrielle et al., 2009; Pérez et al., 2021) and/or the closing of the eastern Tethys gateway (e.g., Hamon et al., 2013; Steinthorsdottir et al., 2020; Woodruff and Savin, 1989). However, large uncertainties in the timing of these ocean gateway changes, which may have affected Southern Ocean bottom waters and Antarctic ice volume to different extents, hamper an unambiguous correlation. Overall throughout the middle to late Miocene, climate modelling indicates that intermediate to deep waters in the Southern Hemisphere may have been warmer than modern due to differences in ocean currents related to the open Central American Seaway (e.g., Burls et al., 2021). Although this does not immediately help explain the sequence of events observed in our record during the MMCT, it may at least shed some light on the elevated temperatures during the MMCO and the rebound to warmer temperatures after the observed cooling at the MMCT.

Given the lack of similar data from a range of locations and water depths, it is difficult to assess how widespread the observed deep ocean cooling was, and whether the cooling reflects variations in the properties of a single bottom water mass or rather shifts in the boundaries between different water masses. Results from a climate modelling study indicate spatially heterogeneous temperature changes in large parts of the Southern Ocean during the MMCT, caused by a complex interplay between winds, ocean circulation and sea ice (Knorr and Lohmann, 2014). Nevertheless, the similarity between the early MMCT BWT decreases observed at Site 747 in the Southern Ocean and at Site 806 in the deep tropical Pacific (Lear et al., 2015; Fig. 3) suggests that the temperature signal was transferred from the Southern Ocean region covered by our Site 747 record into the Pacific Ocean basin. This interpretation may imply deep water formation in the Southern Ocean and an ocean

gateway configuration similar to today, with an active Antarctic Circumpolar Current and continuous export of deep ocean water masses formed in the Southern Ocean to lower latitudes.

Compared to the reconstructed early bottom water cooling, the warming during the later phase of the MMCT starting at or just after the stepped main increase in benthic $\delta^{18}$O (~13.9–13.7 Ma) is even more enigmatic. It could signify a return to the circulation state before the early MMCT bottom water cooling. Alternatively, substantial ice expansion could have led to increased stratification and shielding of deeper waters in the Southern Ocean, resulting in a warming of these water masses. Majewski and Bohaty (2010) measured $\delta^{18}$O on middle Miocene benthic (*Cibicidoides* spp.) and planktic foraminifera (e.g.,

*Globigerina bulloides*) at Site 747 across the MMCT. These authors documented a marked increase in the calculated $\delta^{18}$O differences between *Cibicidoides* spp. and *G. bulloides* (vertical $\delta^{18}$O gradient) during the main increase in $\delta^{18}$O and interpreted this signal as a surface freshening. A freshening in the upper waters of the open Southern Ocean may be related to an increase in meltwater input from a growing ice sheet on Antarctica and possibly the melting of northward-exported sea ice (e.g., Crampton et al., 2016; Sangiorgi et al., 2018; Sigman et al., 2004). An upper ocean freshening across the MMCT

was also reconstructed at Site 1171 (Leutert et al., 2020; Shevenell et al., 2004). At high southern latitudes, salinity has a large effect on stratification (e.g., Kuhnert et al., 2009). We hypothesize that a Southern Ocean freshening concurrent with Antarctic ice sheet expansion may have decreased convective vertical mixing resulting in a shielding of upper ocean waters from comparably warm deeper waters. This stratification mechanism may have influenced Southern Ocean BWTs during the late MMCT, explaining the transient bottom water warming and the different trends of upper ocean temperature and BWT.

An increase in stratification starting between 14 Ma and 13.5 Ma is also supported by an increase in dissolution at that time (Figs. 4 and S9), which may be related to reduced ventilation and an increase in $CO_2$ storage in the deep ocean. A similar mechanism may, in principle, have acted in the opposite direction during the earlier cooling.

Further clues can be obtained from the evolution of $\delta^{18}O_{bw}$, which we can calculate from measured benthic foraminiferal $\delta^{18}$O in combination with $\Delta_{47}$-based BWTs (Fig. 5e). Due to the comparably large random errors in our $\Delta_{47}$-based BWT

estimates, the propagated uncertainties in $\delta^{18}O_{bw}$ are also large. In addition, the foraminiferal $\delta^{18}$O values used in the calculations could also include an ocean pH component (Zeebe, 1999) and/or reflect foraminiferal species-specific effects on $^{18}$O fractionation that were different than those included in existing calibrations (e.g., Bemis et al., 1998; Marchitto et al., 2014). However, other systematic biases in our $\delta^{18}O_{bw}$ estimates may be smaller compared to alternative methods (e.g., paired benthic foraminiferal Mg/Ca and $\delta^{18}$O measurements) because $\Delta_{47}$ signatures seem to be insensitive to foraminiferal

species-specific vital effects and environmental parameters other than temperature (e.g., Leutert et al., 2019; Peral et al., 2018; Piasecki et al., 2019; Tripati et al., 2015; Watkins and Hunt, 2015).

At Site 747, the MCO is characterized by variable $\delta^{18}O_{bw}$ values ranging from around –0.1 ‰ to 0.7 ‰ (~16.0–14.4 Ma; Fig. 5e). For the cold BWT period during the MMCT, $\delta^{18}O_{bw}$ is overall lower than before with values from around –0.3 to 0.1 ‰ (~14.4–13.6 Ma), followed by comparably high post-MMCT values of ~0.7–1.0 ‰ (~13.6–12.2 Ma). All

reconstructed $\delta^{18}O_{bw}$ values are consistently higher than expected for minimal ice (i.e. –0.89 ‰ according to Cramer et al. (2011)). Overall, our $\delta^{18}O_{bw}$ values from Site 747 correspond well to those reconstructed with a similar approach at Site 761 (Modestou et al., 2020; Fig. 5) and those based on Mg/Ca BWTs at Site 806 (Lear et al., 2015). Thus, mounting evidence from various sites and proxies suggests that high $\delta^{18}O_{bw}$ values represent a robust feature of the middle Miocene. Taken at face value, these results suggest the presence of substantial ice sheets primarily on Antarctica and possibly also on Greenland (e.g., Thiede et al., 2011) in times of warm bottom waters (e.g., Lear et al., 2015; Modestou et al., 2020). Short-lived (orbital-scale) minima in global ice volume during peak MCO interglacials (e.g., Levy et al., 2016) may not be visible in the $\Delta_{47}$-based records from Sites 747 and 761 due to their temporal resolution and possible averaging over glacial and interglacial climate states. In addition to the extent of global ice volume, however, the $\delta^{18}O_{bw}$ may also reflect an Antarctic ice sheet oxygen isotopic composition that was different from today (Langebroek et al., 2010) and/or variations in deep ocean salinity (e.g., Modestou et al., 2020). At Site 747, the latter seems especially likely during the cold BWT period in the MMCT where $\delta^{18}O_{bw}$ is low (~14.4–13.6 Ma), possibly reflecting a cold and fresh water mass bathing the site.

The increase in $\delta^{18}O_{bw}$ after the cold BWT period likely includes both a salinity and ice volume component, given that it occurs close in time to the main stepped benthic $\delta^{18}O$ increase starting between 13.8 Ma and 14.0 Ma (Fig. 5d). This marked feature of the MMCT may reflect an increase in Antarctic ice volume during a prolonged period of low seasonal contrast over Antarctica (declining eccentricity, decreasing amplitude variations in obliquity, Fig. 5a), as pointed out by Holbourn et al. (2005). The inferred ice volume increase is supported by ice-rafted detritus records from two study sites offshore East Antarctica, Wilkes Land IODP Site U1356 and Prydz Bay ODP Site 1165 (Pierce et al., 2017), as well as multiproxy evidence for an episode of maximum ice sheet advance (MISA-4) recorded in the ANDRILL (AND)-2A drill core from the western Ross Sea, Antarctica (Levy et al., 2016). Similarly, an earlier period of maximum ice sheet advance documented in the Ross Sea around 14.7–14.6 Ma (MISA-3) corresponds to a maximum in $\delta^{18}O_{bw}$ at Site 747 suggesting larger global ice volume. Unfortunately, the section from 14.4 Ma to 13.8 Ma is missing in the AND-2A core (Levy et al., 2016) preventing us from a final assessment to what extent the minimum in $\delta^{18}O_{bw}$ reflects a substantial global ice volume minimum, or rather another factor such as low salinity (as discussed above). More proxy environmental data from Antarctica and its continental shelves as well as additional BWT and $\delta^{18}O_{bw}$ records from different sites and water depths in the Southern Ocean will allow for a better understanding of the intriguing features of the MMCT recorded at Site 747.

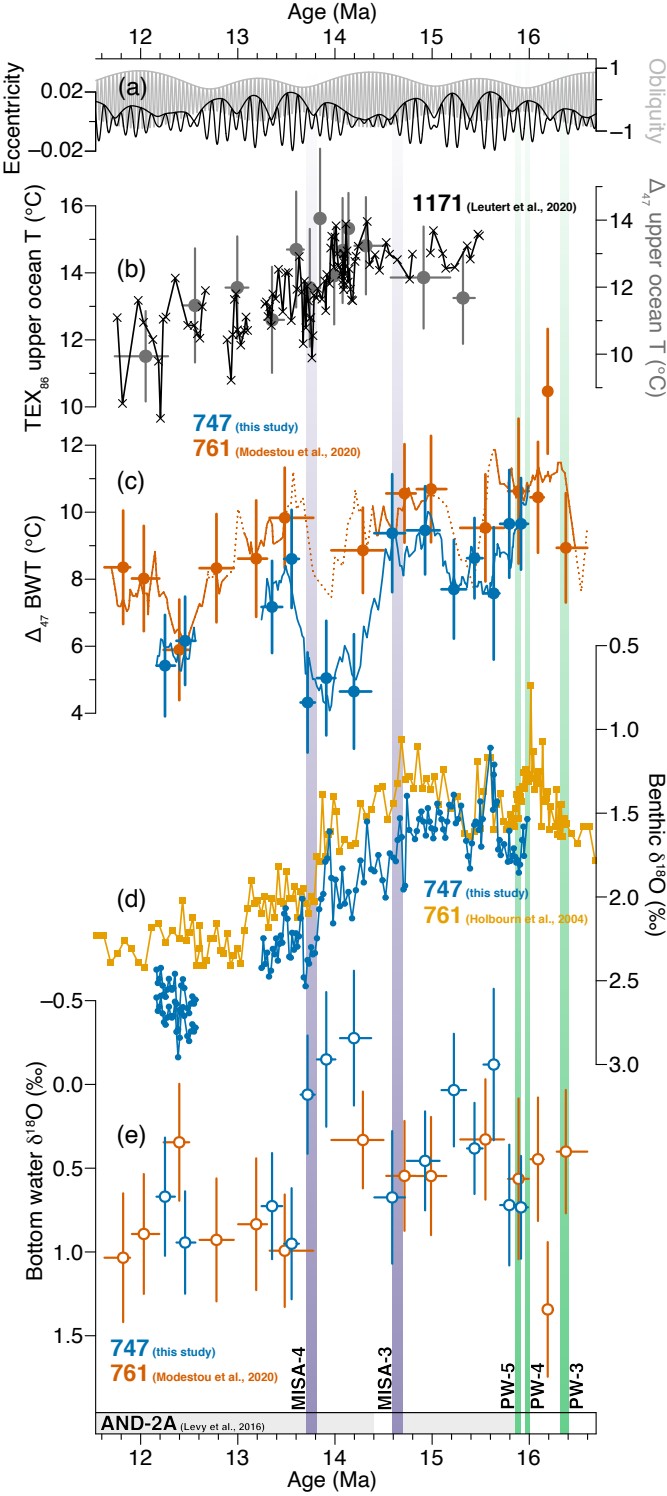

**Fig. 5:** Compilation of records for the MMCT. (**a**) Filter of obliquity centered at the 40 kyr-periodicity with its amplitude modulation (light gray) and filter of eccentricity centered at the 110 kyr-periodicity with its amplitude modulation (black), (**b**) $\Delta_{47}$- and $TEX_{86}$-based upper ocean temperatures from ODP Site 1171 on the South Tasman Rise are shown with (**c**) $\Delta_{47}$-based bottom water temperatures (BWTs), (**d**) benthic foraminiferal $\delta^{18}O$ and (**e**) bottom water $\delta^{18}O$ ($\delta^{18}O_{bw}$) from ODP Sites 747 and 761. In addition, we highlight distinct episodes of maximum ice sheet advance (MISA-3 and MISA-4, purple bars) and peak warmth (PW-3 to PW-5, green bars) around Antarctica derived from the ANDRILL (AND)-2A drill core (western Ross Sea); missing sections in AND-2A are light gray shaded (Levy et al., 2016). $\Delta_{47}$-based BWTs (Modestou et al., 2020; this study) and upper ocean temperatures (Leutert et al., 2020) are shown with 68 % confidence intervals. These upper ocean temperatures were derived from *G. bulloides* that are assumed to dwell around 200 m water depth in the Southern Ocean (Vázquez Riveiros et al., 2016). $TEX_{86}$-based temperatures (Leutert et al., 2020) are based on the subsurface calibration of Ho and Laepple (2016). Site 761 benthic $\delta^{18}O$ values are from Holbourn et al. (2004). Orbital parameters are from Laskar et al. (2004). Using the software AnalySeries 2.0.8 (Paillard et al., 1996), we applied Gaussian band-pass filters centred at wavelengths of 40 kyr (frequency: 0.025 kyr$^{-1}$, bandwidth: 0.002 kyr$^{-1}$) and 110 kyr (frequency: 0.009 kyr$^{-1}$, bandwidth: 0.003 kyr$^{-1}$) to obliquity and eccentricity, respectively (see also Fig. S10 for orbital parameters).

# 5 Conclusions

We constrain the middle Miocene BWT evolution at Site 747 in the Southern Ocean with clumped isotope thermometry. Similar to existing BWT reconstructions from lower latitude sites, we find that Southern Ocean BWTs were substantially warmer than today, despite the presence of ice sheets on Antarctica. The discrepancies between $\Delta_{47}$- and Mg/Ca-based BWTs observed at Site 747 may be caused by changes in deep water carbonate ion saturation, but further Mg/Ca and $\Delta_{47}$ measurements are needed to conclusively test this hypothesis. We cannot fully rule out a dissolution effect on benthic foraminiferal $\Delta_{47}$, although there is currently no evidence for such an effect. Taken at face value, our $\Delta_{47}$ values indicate pronounced shifts in Southern Ocean BWTs, which resemble observations at equatorial Pacific Site 806. We observe a substantial BWT decrease of ~3–5°C during the early MMCT that was followed by a transitional smaller warming and an eventual return to cooler conditions. The reconstructed changes in BWT and $\delta^{18}O_{bw}$ indicate a more complicated sequence of events surrounding the MMCT than previously appreciated based on benthic $\delta^{18}O$ alone. These findings suggest the involvement of additional feedbacks and thresholds in middle Miocene ice growth and possibly regional effects, for example caused by a reorganization of the water mass structure, on BWT and $\delta^{18}O_{bw}$ at Site 747. We hypothesize that an important factor could be shifts in the vertical density structure of the Southern Ocean. The reconstructed BWTs may in part reflect changes in heat transport between upper and deep ocean, induced by growing ice sheets on Antarctica. Independent higher-resolution BWT records from further locations in and outside the Southern Ocean would enable examination of the spatial scale of the changes observed at Site 747 as a basis for better understanding the drivers of the MMCT.

## Appendix A: Clumped isotope methodological details

Clumped isotope data are presented in the conventional $\Delta_{47}$ notation, which is defined as follows (e.g., Eiler, 2007; Huntington et al., 2009):

$$\Delta_{47}\ (\text{‰})\ =\ [(\frac{R^{47}}{R^{47*}} - 1) - (\frac{R^{46}}{R^{46*}} - 1) - (\frac{R^{45}}{R^{45*}} - 1)] \times 1000 \qquad (A1)$$

$R^i$ are the measured abundance ratios of mass i relative to mass 44. $R^{i*}$ represent the stochastic abundance ratios calculated from the bulk isotope composition of the sample ($\delta^{18}O$ and $\delta^{13}C$).

All (clumped) isotope measurements (see Tables S1 and S5) were carried out in micro-volume mode. At the University of Bergen (UiB), we followed the long-integration dual-inlet (LIDI) protocol (Hu et al., 2014; Müller et al., 2017), whereas the measurements at ETH Zurich were performed via repeated cycles of alternating reference and sample gas measurements
(Meckler et al., 2014; Rodríguez-Sanz et al., 2017). For data processing, we used the community software "Easotope" (John and Bowen, 2016). The different steps for calculating the final $\Delta_{47}$ values include a pressure-sensitive baseline correction (Bernasconi et al., 2013; He et al., 2012; Meckler et al., 2014) and a conversion into the absolute reference frame (Dennis et al., 2011). For the conversion into the absolute reference frame, we utilized replicate measurements of three (UiB) respectively four (ETH Zurich) different correction standards from a window of ±12–40 standards around the sample
replicate. At UiB, we used the carbonate standards ETH-1, ETH-3 and ETH-4 for correction from October 2016 to December 2016; ETH-2 was used for monitoring during this interval. From August 2018 to June 2019, ETH-1, ETH-2 and ETH-3 were used for correction and ETH-4 for monitoring. For the measurements carried out at ETH Zurich, ETH-1, ETH-2, ETH-3 and ETH-4 were all included in the correction procedure. The accepted ETH standard values are from Bernasconi et al. (2018). These ETH standard values were determined using an acid fractionation correction of +0.062 ‰ (Defliese et
al., 2015). Measured $\delta^{18}O$ and $\delta^{13}C$ values were drift-corrected based on three (UiB) respectively four (ETH Zurich) different correction standards (with scale "stretching" only applied for $\delta^{18}O$ at UiB and for both $\delta^{18}O$ and $\delta^{13}C$ at ETH). All isotope data were calculated with the Brand correction parameters (Daëron et al., 2016). Further details on analytical and data processing methods can be found elsewhere (Leutert et al., 2019; Piasecki et al., 2019).

For temperature error propagation, the Meinicke et al. (2020) calibration dataset was used to calculate variances of
calibration slope and intercept as well as the covariance of calibration slope and intercept. We note that the covariance of calibration slope and intercept is required for error estimation as the errors in slope and intercept of the calibration line are correlated. Then, the variance-covariance matrix with these values was used to propagate calibration and measurement errors (similar as described in the supporting information of Huntington et al. (2009)) following conventional error propagation

procedure. As pointed out in previous studies (e.g., Huntington et al., 2009; Peral et al., 2018), the calibration error in
clumped isotope temperature estimates was observed to be very small compared to analytical uncertainties in this study.

We excluded three clumped isotope measurements as outliers, based on their offset of more than four standard deviations
(4×0.037 ‰, estimated from the long-term mean reproducibility of all standards) from the mean. Raw standard and sample
measurement data are included in the supplement and will be available on EarthChem at the time of publication.

**Data availability**

The data from this paper are archived in the supplement. In addition, the final temperature data will be published at Pangaea
(https://doi.pangaea.de/10.1594/PANGAEA.923258)    and    the    full    raw    data    on    the    EarthChem    Database
(https://doi.org/10.26022/IEDA/111808).

**Author contribution**

T.J.L. and A.N.M. initiated and designed the study. T.J.L. generated and analysed clumped isotope data under the oversight
of A.N.M., S.M. and S.M.B. All the authors contributed to the palaeoceanographic interpretation. T.J.L. wrote the paper with
contributions from A.N.M., S.M. and S.M.B.

**Competing interests**

The authors declare that they have no conflict of interest.

**Acknowledgements**

We thank Enver Alagoz and Inigo Müller for analytical support, Janika Jöhnck for insightful discussions and all authors who
shared their published data. This research used data and samples provided by the Ocean Drilling Program (ODP) and the
International Ocean Discovery Program (IODP), sponsored by the US National Science Foundation (NSF) and participating
countries. Funding for the research was provided by the European Research Council (ERC) under the European Union's
Horizon 2020 research and innovation programme (grant agreement No 638467) and by the Trond Mohn Foundation.

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
