# Peer review of "Southern Ocean bottom water cooling and ice sheet expansion during the middle Miocene climate transition"

_Climate of the Past, 2020_

## Referee Comment (RC1) · Anonymous Referee #1 · 22 Jan 2021

General comments:

Leutert et al present an interesting new record of bottom water temperatures from the Kerguelen Plateau during the middle Miocene – a time of substantial ice-sheet growth and cooling. The record will be a valuable contribution to our understanding of ice volume vs temperature changes in this interval. A revised age model for ODP Site 747 is presented and seems to be robust. New benthic stable isotope data match well with existing records. The paper is overall well-written; however, I suggest a substantial overhaul of the discussion.

The stand-out feature of the new temperature record is a large, transient ( $\sim$ 0.8 Myrlong) cooling of 3-5°C during the middle Miocene climatic transition, between  $\sim$ 14.5 and 13.7 Ma. The fact that cool temperatures are recorded in three consecutive intervals (each made up of  $\sim$ 30 analyses) suggests it is a robust signal. Because this large cooling occurs during an interval with only a small increase in benthic  $\delta$ 180, the implication is that it was accompanied by significant de-glaciation lasting  $\sim 0.8$  Myr (shown by the large decrease in bottom water  $\delta$ 18O). This aspect of the record (its plausibility and implications, possible mechanisms that might have caused it, whether there is any other evidence for deglaciation at this time) are not discussed in enough detail in the paper. For example, the large step decrease in bottom water  $\delta$ 180 at ~14.5 Ma is barely mentioned. No clear explanation for the cooling is given (although the subsequent warming is discussed). There is very little discussion of bottom/intermediate water circulation, which water masses might have bathed the site and how this might have changed over the study interval, deep-water formation (e.g. proposed Miocene onset of Antarctic Bottom Water Formation in the Weddell Sea, Pérez et al., 2020), changes in Antarctic gateways, etc. that may have influenced the temperature record. Importantly, the reader does not know what the Miocene paleodepth of the site was and to what extent benthic forams at this site might record local versus global temperature signals. The bottom-water temperature trends at Site 747 (based on  $\Delta$ 47) are quite similar to those seen at Site 806 based on Mg/Ca but not other sites, which is really interesting. Is there a water mass/circulation-related explanation for this?

**Specific comments:**

I have a couple of suggestions to improve Figure 1: Firstly, I would use a different (more inclusive) colour scale for the temperature map, as the rainbow colour scale is now widely known to be a poor choice both for colour-blind people and also for reproduction in grayscale. Secondly, I find the plate tectonic reconstruction shown in this figure difficult to interpret, because it shows tectonic plates including ridges and continental shelves, rather than a land-sea mask or reconstructed bathymetry. I suggest that the authors use instead a paleogeographic map which
would more clearly show the distribution of continents and oceans and the paleodepths of sites; e.g. the Scotese paleogeographic reconstruction maps (Paleomap project); Straume et al. 2020 (paleobathymetry reconstructions available at 1 Ma resolution: https://zenodo.org/record/4193576#.YAb\_heB7IXh); or Cai et al 2017 (which includes digital global paleogeographic maps in the supplement, including a 14 Ma reconstruction).

**Introduction**

"The middle Miocene geographic position of Site 747 relative to Antarctica was similar to today"; I found this statement a bit lacking in detail on paleolatitude, setting, etc., so I suggest expanding on this. Also the paleodepth of the site is not discussed – could a shallower paleodepth contribute to the relatively warm temperatures you reconstruct compared to modern, and the relatively large changes?

The  $\Delta$ 47 temperature proxy is well introduced, however given that you list all the potential caveats of the Mg/Ca paleothermometer as applied to benthic foraminifera, I feel the  $\Delta$ 47 proxy gets off quite lightly. A brief summary of the potential impact of diagenesis (dissolution, recrystallization, and overgrowth), burial, or other known non-thermal processes on  $\Delta$ 47 in benthic foraminifera and their effect on reconstructed temperatures would be useful, even though you discuss this in detail later.

Methods/Results & Discussion: I think it would be clearer if the Results and Discussion were separated.

Age model: I would move the Age Model section up so that it follows the Site Details section. In addition, an age-depth plot for Site 747 (in the supplement if necessary) showing all of the different tie points used (magnetostratigraphy, isotope-based, biostratigraphy) and the described hiatus would be very useful. Is the assumption that Site 806 sedimentation rates were constant and similar either side of the orbitally-tuned record between 14.1 and 13.3 Ma supported by shipboard magnetostratigraphic and biostratigraphic datums? I would verify this if you have not already, especially given

CPD
that this is the record that has the most similar trends to your new record. With this assumption, the comparison is not very robust. Presumably the original publication of the Mg/Ca record had age constraints that covered the whole interval? The calculation of uncertainties should be briefly described, rather than just referring to the supplement of another paper.

"Results from adjacent samples are pooled to achieve this number of measurements" Please be more precise about how many adjacent 2-cm samples were pooled together (mean, min, max depth/age intervals over which results were averaged). Samples were run on two different machines, but as far as I can see we cannot tell from the figures which data were run on which machine. It might be useful to colour code data points in Figure S3 to show that there are no machine offsets. Are the cited external reproducibilities for both of these machines? As a side note, I feel like Figure S3a should be shown in the main text (maybe as a top panel in Fig. 3), as it shows the raw data upon which all your subsequent data averaging and interpretations are based.

Fig. 3: horizontal solid lines: averaging intervals; it is not clear to me why the points are not plotted in the middle of the averaging intervals. Is the age of the points weighted towards the highest data density? Why was a 400-kyr moving window approach used rather than a Gaussian-Weighted Filtering approach, as in Modestou et al 2020? I am not sure which method is most appropriate, but the Gaussian-Weighted Filtering approach does seem to smooth out the small-scale features noted by the authors to be caused by scatter in measurements. Add an error bar for Mg/ca-based temperatures. On Figures 2 and 3, it would be helpful to highlight the middle Miocene climatic optimum and transition intervals, and also the hiatus.

Line 192 – again please specify how large/variable the intervals over which data were averaged are in the text. "We note that small-scale features in the moving average curves are likely caused by the scatter in the underlying individual  $\Delta$ 47 measurements, and should not be interpreted as real climate signals" For clarity, please quantify small-scale (

sion): I suggest citing temperature confidence intervals ( $\pm x^{\circ}C$  at x CI) when describing absolute values, this will help to emphasise which trends are significant given the large error bars on  $\Delta 47$  temperatures (e.g. a 3-5°C cooling is larger than 68% CI). Line 218: How do the recalculated bottom-water temperatures from Site 761 compare to the originally published values? Line 229: What artefacts could result from comparing a low-resolution record of discrete samples (each representing maybe 1-2000 years, without knowing if it is a glacial or an interglacial) with a record where each sample integrates hundreds of thousands of years? Line 269: do the authors have any suggestions as to how to investigate this? Line 288: include d18Obw errors in the text. "For the later MCO (15.6–13.9 Ma), our estimates of  $\delta$ 18Obw range from around -0.3 ‰ to 0.7 ‰5 This statement doesn't really adequately describe the large step changes in reconstructed bottom water  $\delta$ 18Obw at ~14.5 Ma and 13.7 Ma. Line 294: due to their temporal resolution and also due to averaging of many samples probably mixing glacial and interglacial climate states. Line 326: what was the interpretation of this change in vertical gradient?

Reference:

Pérez, Lara F., Yasmina M. Martos, Marga García, Michael E. Weber, Maureen E. Raymo, Trevor Williams, Fernando Bohoyo et al. "Miocene to present oceanographic variability in the Scotia Sea and Antarctic ice sheets dynamics: Insight from revised seismic-stratigraphy following IODP Expedition 382." Earth and Planetary Science Letters 553 (2020): 116657.

---

## Referee Comment (RC2) · Anonymous Referee #2 · 5 Mar 2021

Leutert et al., present the first high-latitude independent bottom water temperature records for the mid Miocene, spanning the bulk of the Miocene climatic optimum and the Miocene climate transition. By using clumped isotope thermometry, the authors circumvent known issues that affect more traditional BWT proxies, such as Mg/Ca.

The main contributions of this manuscript are twofold: 1) By providing independent mid Miocene BWT records, they can evaluate the reliability of deep sea benthic foraminiferal D47 and Mg/Ca records as a BWT proxy. This comparison confirms that D47 is likely an independent temperature proxy that predominantly records BWT, whereas Mg/Ca is affected by non-thermal effects. 2) They show that the main trends

in mid Miocene BWT, as reconstructed by D47, are observed at both high and low latitude sites, but are somewhat decoupled from the main trends in ice growth across the mmct. They speculate that regional freshening in the upper water column may be a mechanism to explain this decoupling.

My main concerns are to do with the organization of the manuscript (I have made more specific comments about this below), which can be easily address: - The authors have spread methodological information across the methods, results/discussion and appendix. I found this confusing and will make it hard for readers to later find their methodological approaches. - I found it especially distracting to have Section 3.1 and 3.2 jump between methods, results and discussion. If I were coming back to this paper to find either methodological information or reread the scientific discussion, I would find this frustrating. I think the manuscript would be clearer if the authors could reorganize and group this information better. - The authors also discuss supplementary figures in quite a lot of detail in the main text, so I don't understand why some of those figures aren't incorporated. I am fine with supplementary figures, if the main discussion of those figures is also in the supplement. I found it frustrating to have to go back and forth between the main text & supplement where SI figures were being discussed in detail in the main text. In some cases, the supplementary figures are also only slightly more expanded versions of the main figures, so I don't understand why the supplementary version isn't used instead of the current main version.

Structure concerns aside, the manuscript presents one of the few high-latitude deep sea temperature records of the mid Miocene. I think the manuscript represents a good contribution to Climate of the Past, with interesting implications for both Mg/Ca thermometry and mid Miocene climate reconstructions.

Suggestions in order of appearance:

- Page 1 -

Ln 16: Could you specify which other regions/sites you compare to?

Ln 28: In my experience the mmct is defined as the specific benthic isotope excursion ~13.9 Ma (eg Holbourn et al., 2005), much like you'd recognize CM6 or CM5a/b. Could the authors provide a reference for where their definition comes from?

- Page 2 -

Ln 32: Can the authors include the original publications that produced $CO_2$ records across the mmct, for instance Foster et al., 2012 EPSL?

Lns 39-41: I don't think the Fairbanks equation is appropriate here, considering that the isotopic composition of the Miocene ice itself was likely different. Can values reported in more recent modelling studies (specifically Gasson et al., 2016, PNAS) possibly provide a better estimate of this?

- Page 3 -

Lns 74-75: Can you introduce here what you mean by low data density? Are you hoping to track orbital-scale (eccentricity, obliquity, precession) variability over this time period? This may become apparent later on, but it would be good to introduce this more clearly here, as low vs high data density can mean very different things to different people.

Ln 77: "calcites" should be "calcite".

Ln 78 / Ln 83: Please specify that this is in the Indian sector of the Southern Ocean.

Ln 88: It would be useful to incorporate the target sample resolution earlier on, especially if you can link the temporal sampling resolution to your goal (still at this stage not clear whether you aim to just get a grasp of long-term changes, or also want to pick up orbital scale variability).

Lns 88-90: Can you clarify this statement? Do you mean you are using a composite depth scale? Based on the supplementary tables, the authors use a revised mbsf (rmbsf), but they don't actually define that anywhere. It's great to see the authors include the full sample ID and depth, but if they could additionally include the original

mbsf as a column in Table S5, that would be better, especially as they seem to refer to the mbsf not rmbsf depths in Section 2.1.

- Page 4 -

Figure 1: - It took me a while to understand this figure, and especially to understand that the two maps are of the same area. I was confused as 747 is only highlighted on the left map & 761 only on the right map. Could you potentially adapt this figure to show the full map? Or annotate more clearly on the figure that 1700 m / 2200 m depth are the modern 747/761 water depths respectively? - Also, could the authors add 1171, and ideally U1335/7/8 on the map? I understand the eastern equatorial Pacific sites are hard to fit on with the globe shown as is, but 1171 can definitely be added.

Lns 106-107: There is some evidence that these species can have different d13C signatures, although this isn't often seen. Has there been any research on clumped isotopes being comparable between the two species? Did you measure the species separately in any of your samples to check they are comparable? Later comment: much of this is later included in the results/discussion section. I found that very confusing and would recommend the authors address interspecies offsets in d18O/d13C/D47 in the methods.

- Page 5 -

Ln 110: Rinsed in DI water?

Section 2.3 vs Appendix A: As D47 is a key contribution here, I found it confusing that the clumped isotope methods were split between this section and Appendix A.

- Page 6 -

Lns 143-145: Can the authors provide these recalibrated records in their supplementary data for community use, of course appropriately referencing the original studies? This would greatly help update those records to this more recent calibration. From the supplementary information, it seems the 761 data is included, but not the 1171 data.

[Figure]

Lns 150-151: GTS2020 was recently published (Raffi et al., 2020), so I would recommend updating the magnetostratigraphic tie points to the most recent timescale. This may not make much difference for the younger interval, but for the oldest reversals used, GTS2020 uses the Chrons recalibrated by Kochhann et al., 2016. As the authors use the Kochhann ages for the d13C based ties, for consistencies sake, I would recommend using the Kochhann et al., 2016/GTS2020 ages for the magnetostratigraphic tie points as well.

Lns 167-168: Ah, the authors discuss interspecies offsets here. I think it would be helpful to discuss in the methods that both species were measured in the same sample to quantify interspecies offsets.

- Page 8 -

Figure 2: The authors have done a good job at compiling stable isotope records from different regions; however, I think they are missing some data in the youngest interval. Nathan and Leckie, 2009 (Palaeo3) published low-resolution benthic stable isotope data at 806 and Tian et al., 2017 (Gcubed) and 2018 (EPSL) provides benthic stable isotope data from U1337 between 0 and 16 Ma. Including relevant parts of these datasets could benefit their inter-regional comparison, as they hardly show any data younger than 12.7 Ma in the eastern/western equatorial Pacific Ocean.

- Page 9 -

Section 3.2 title: I'm not sure the title really reflects the content of the section. If the authors do reorganize their current section 3 to separate out methods, results and discussion, subsections of this topic could be helpful.

Ln 189: I appreciate the averaged D47 signal is the one the authors want to take forward in their discussion, but can they incorporate Figure S3a into the main manuscript? The D47 data is a key result of the study and they refer to Figure S3 a few times, so it would be better for the reader not to have to flip back and forth between the main text

and supplement so often.

First two paragraphs of 3.2 (Lns 190-199 & 200-204): I found the way methods-results-discussion were all mixed together in section 3 to be confusing. Can the authors re-structure it, so the information is easier to find? I found it distracting to continually jump back and forth between methodological information & scientific discussion. The first paragraph is very methodological (190-199), followed by a results paragraph (200-204), before the authors move on to their discussion. If I were rereading this paper at a later stage for their climatic interpretation and discussion, I would find the inclusion of methodological information in the discussion distracting.

- Page 10 -

Figure 3: I appreciate this is not data produced by the authors, but if they could provide error estimates for the Mg/Ca BWT, that would be helpful.

- Page 11 -

Lns 220-222: I don't fully understand what the authors are trying to say here. It currently reads like the authors are saying that Miocene BWT at 747/761 were similar to the present day, but then in the following sentence, they say Miocene BWT were considerably warmer. Do they mean the difference between 747 and 761 is similar to the present day?

Lns 229 - 231: Showing these BWT records overlaid would help illustrate this point.

Lns 234-245: Can the authors integrate Figure 3 with Figure S4? The authors discussed the various Mg/Ca records included in Figure S4 quite a lot in this paragraph, but only include 2 in Figure 3. I again found it frustrating to have to switch between the main text and supplement to look at data that was discussed in depth in the main text.

- Page 12 -

Ln 257/Figure 4: I don't understand why Figure S5 isn't used in the main manuscript

instead of Figure 4, especially as the inclusion of CO32- data is the main difference between the two figures.

Lns 257-258: This sentence seemed out of place, as they only discuss 747 in this paragraph and only briefly mention 1171 in the previous paragraph.

- Page 14 -

Lns 280-281: This information feels more appropriate for the methods.

Ln 284: "complicating" should be "complicated".

Lns 291-293: What could past differences in the isotopic composition of the Miocene ice sheet mean for these estimates? The authors mention absolute estimates in the introduction, so it seems odd that they do not come back to absolute estimates here after reconstructing d18Osw. If the authors considered the potential past isotopic composition of the ice sheet used in Gasson et al., 2016, they could transfer their new d18Osw estimates to absolute sea level estimates. I think this could be worth including, especially as they mention absolute estimates in the introduction.

Lns 293-294: This sentence isn't entirely clear; can you rephrase it? Also, could you include the equations used in Figure S6 in the caption or figure itself?

Ln 298: Can the authors rephrase the part of the sentence here to do with orbital parameters? When I first read it, it seems to be undermined by the previous sentence where the authors say that they can't be sure they pick up orbital-scale minima in global ice volume due to the temporal resolution of the D47 records. If they can make it clearer that they mean to look at the influence of longer-term cycles in orbital parameters (e.g., 2.4 Myr/1.2 Myr amplitude modulations), rather than shorter-term orbital cyclicity, that would be helpful.

- Page 15 -

Figure 5: - It was hard to read all the details in this figure. Could it be made bigger/wider? - Also, I appreciate the authors are not trying to discuss orbital scale variability, but it could be useful to include one of the higher resolution benthic stratigraphies as a reference for the exact timing of different events. A composite of the records shown in Figure 2 could be used, or the authors could include the new astronomically tuned "zachos curve" from Westerhold et al., 2020. - In panel A, could the authors highlight the amplitude modulation of the longer-term eccentricity and obliquity cycles (for instance, as done in Liebrand et al., 2017 PNAS)? - In the caption, it's unclear whether the authors mean that the "upper ocean temperatures" at 1171 are a shallow bottom water temperature, or a (near)-surface ocean signal, without being aware of the original study.

- Page 15 -

Lns 319-320: The main d18O decrease certainly occurs after the big drop in temperatures, but I think it also looks like the 747, 761 d18O record is also increasing between the two purple bars, which might indicate that the decoupling is more nuanced than the authors initially suggest.

Ln 329: Could you include the location of 1171 on Figure 1.

Lns 333-335: Could the authors provide a bit more mechanistic detail here about why expansion of the Antarctic ice sheet would result in a freshening of the upper water column? A greater uptake of fresh water in an ice sheet could arguably increase local salinity, not cause freshening. Further explanation might help explain the link the authors suggest.

---

## Referee Comment (RC3) · Anonymous Referee #3 · 19 Mar 2021

In their study Leutert and colleagues present a record of bottom water changes from ODP Site 747 spanning 16.0-12.2 Ma. The data and the integration of the records to existing geological data reveals fascinating insights into the transition between the Middle and Late Miocene (MMCT). However, in the present manuscript version the authors are retentive in the presentation and discussion of their results. Therefore, it is strongly recommended to further exploit the potential of the study.

Comments:

- Focus of the study is the MMCT. Leutert et al. define this interval as ∼14.5-13.0
Ma, as it contains key changes in the presented records. Given the relatively long definition of this interval it is recommended to define a nomenclature and time intervals for different sub-phases along the key bottom water temperature (BWT) and bottom water d18O (BWd18O) changes. Once introduced they should be used consistently throughout the paper. For instance, statements like 'Our dataset demonstrates that BWTs at Site 747 decreased by ∼3–5°C across the MMCT.' don't do justice to the complexity of the recorded changes, since the full cooling magnitude can be already reached between ∼14.5-14.3 Ma, i.e. even before the phase of major ice growth.

- The sub-intervals might be chosen to cover the divergence of BWT and BWd18O, including a rapid cooling and abrupt warming into and out-off the phase of minimum BWT (between ∼14.3. 13.7 Ma). This phase is accompanied by a pronounced decrease (increase) in BWd18O at the beginning (end) of this interval. So far, the focus is towards the end of this interval.

- In the current manuscript version, the timing between BWT/ BWd18O changes to upper ocean temperature changes is touched only marginally. However, the timing of these changes can be a key to better differentiate between various forcing mechanisms.

- Although this paper is a data study, it would be helpful to relate a growing body of relevant model studies to their findings. Interesting aspects might include e.g. the impact of CO2 or ice sheet changes on upper-ocean and BWT changes across the MMCT. Both factors are expected to have different impacts that can support a mechanistic interpretation, since ice sheet changes might have a more heterogenous impact on these temperature records. In this context Section 11.3.5 (Impact of Ice on Miocene Climate) in the recent review of Steinthorsdotti et al. (2020) (doi.org/10.1029/2020PA004037) might be a helpful starting point.

---

## Author Comment (AC1) · 24 May 2021

Review "Southern Ocean bottom water cooling and ice sheet expansion during the middle Miocene climate transition" by Leutert et al.

**Response to Referee #1**

Please find below the referee's comments in blue font and the authors' response in black font.

General comments:

Leutert et al present an interesting new record of bottom water temperatures from the Kerguelen Plateau during the middle Miocene – a time of substantial ice-sheet growth and cooling. The record will be a valuable contribution to our understanding of ice volume vs temperature changes in this interval. A revised age model for ODP Site 747 is presented and seems to be robust. New benthic stable isotope data match well with existing records. The paper is overall well-written; however, I suggest a substantial overhaul of the discussion.

Reply: We are sincerely grateful for the thoughtful and constructive comments of Referee #1 on our manuscript. Importantly, we will follow the referee's advice and substantially revise the discussion of our new record adding more details about possible climate mechanisms and water circulation during the middle Miocene (see below).

The stand-out feature of the new temperature record is a large, transient (0.8 Myr-long) cooling of 3-5°C during the middle Miocene climatic transition, between ∼14.5 and 13.7 Ma. The fact that cool temperatures are recorded in three consecutive intervals (each made up of ∼30 analyses) suggests it is a robust signal. Because this large cooling occurs during an interval with only a small increase in benthic $\delta^{18}O$, the implication is that it was accompanied by significant de-glaciation lasting ∼0.8 Myr (shown by the large decrease in bottom water $\delta^{18}O$). This aspect of the record (its plausibility and implications, possible mechanisms that might have caused it, whether there is any other evidence for deglaciation at this time) are not discussed in enough detail in the paper. For example, the large step decrease in bottom water $\delta^{18}O$ at ∼14.5 Ma is barely mentioned. No clear explanation for the cooling is given (although the sub-sequent warming is discussed).

Reply: We agree that we have given this early cooling too little attention in the previous version and will discuss it more prominently. We will put forward two possible drivers for

such an early cooling: a relationship to expanding ice sheets or circulation changes in the deep ocean caused by tectonic processes accompanying the opening of Drake Passage and Scotia Sea (e.g., Lagabrielle et al., 2009; Pérez et al., 2021) and/or the closing of the eastern Tethys gateway (e.g., Hamon et al., 2013; Steinthorsdottir et al., 2020; Woodruff and Savin, 1989). However, large uncertainties in the exact timing of these ocean gateway changes, which may have affected Southern Ocean bottom waters and Antarctic ice volume to different extents, hamper an unambiguous correlation. The similar early MMCT cooling observed at Sites 747 (Kerguelen Plateau, Southern Ocean) and 806 (tropical Pacific; Lear et al., 2015) suggests that the Southern Ocean bottom water signal was at least transferred into the Pacific Ocean basin.

We will also extend the discussion of possible mechanisms that may explain the observation of low bottom water $\delta^{18}O$ in times of comparably low BWTs. We propose that a possible alternative interpretation to a transient deglaciation could be a regional bottom water freshening and destratification event, explaining the concurrence of low bottom water $\delta^{18}O$ and low BWT at Site 747, even in the case of only limited deglaciation on Antarctica. The latter is difficult to examine in a conclusive manner here, as the section from 14.4 Ma to 13.8 Ma is missing in the AND-2A core (Levy et al., 2016). Although records from two sites offshore East Antarctica, Wilkes Land IODP Site U1356 and Prydz Bay ODP Site 1165 indicate low ice-rafted detritus values (Pierce et al., 2017) and thus indirectly point to limited ice at that time, more proxy records from Antarctica and its continental shelves as well as additional BWT and $\delta^{18}O_{bw}$ records from different sites and water depths in the Southern Ocean may allow for a better understanding of the features of our Site 747 bottom water record in the future (this will be pointed out in the Discussion of the updated manuscript version).

There is very little discussion of bottom/intermediate water circulation, which water masses might have bathed the site and how this might have changed over the study interval, deep-water formation (e.g. proposed Miocene onset of Antarctic Bottom Water Formation in the Weddell Sea, Pérez et al., 2020), changes in Antarctic gateways, etc. that may have influenced the temperature record.

Reply: See our proposed changes above.

Importantly, the reader does not know what the Miocene paleodepth of the site was and to what extent benthic forams at this site might record local versus global temperature signals.

Reply: The benthic foraminiferal species composition from the middle Miocene sequence from Site 747 is characterized by common deep-sea faunal components and strongly resembles the corresponding middle Miocene sequences from Holes 689B and 690C (Maud Rise), as pointed out by the shipboard scientific party. This evidence supports a lower bathyal to abyssal depth at Site 747 during the middle Miocene (Schlich et al., 1989). We will add a sentence with this information in the main manuscript, but are hesitant to include this more prominently, as we could not find any more precise quantitative estimates of middle Miocene paleodepths for Site 747 and also no evidence for a middle Miocene location of Site 747 in a shallow water environment (e.g., Abrajevitch et al., 2014; Billups and Schrag, 2002; Majewski et al., 2010; Verducci et al., 2009). To the contrary, water depths at Site 747 during the middle Miocene may have even been even larger than today (Schlich et al., 1989). In absence of contrary indications, we interpret Site 747 as recording signals which at the very least are reflective of the Indian Ocean sector of the Southern Ocean, but likely are reflective of processes on a larger scale. In any case, we will point to the uncertainty in the scale of the signal at various positions in the text.

The bottom-water temperature trends at Site 747 (based on $\Delta_{47}$) are quite similar to those seen at Site 806 based on Mg/Ca but not other sites, which is really interesting. Is there a water mass/circulation-related explanation for this?

Reply: The similar BWT patterns reconstructed at Sites 747 (Kerguelen Plateau, Southern Ocean) and 806 (tropical Pacific; Lear et al., 2015) suggests that the Southern Ocean bottom water signal was transferred into the Pacific Ocean basin. This interpretation may imply deep water formation in the Southern Ocean and an ocean gateway configuration similar to today, with an active Antarctic Circumpolar Current and continuous export of deep ocean water masses formed in the Southern Ocean to lower latitudes. This interpretation will be included in our discussion. Possible causes for the differences to other records may include (but are not limited to): Regional differences between water masses bathing these sites, variable pore water chemistry (e.g., bottom water carbonate ion saturation effects on benthic foraminiferal Mg/Ca when measured on epifaunal species as in some of these records), diagenetic effects,

data gaps in proxy records based on only one hole (such as those from ODP Sites 747, 761 and 1171) and aliasing due to low-resolution sampling.

Specific comments:

I have a couple of suggestions to improve Figure 1: Firstly, I would use a different (more inclusive) colour scale for the temperature map, as the rainbow colour scale is now widely known to be a poor choice both for colour-blind people and also for reproduction in grayscale.

Reply: We thank Referee #1 for pointing this out. We will change the colour scale to a colour scale going from blue over white to red. In addition to avoiding rainbow colours scales and the introduction of false perceptual thresholds (e.g., Hawkins, 2015), this type of colour scale also appears to be a better choice for colour-blind people (see "https://colorbrewer2.org").

Secondly, I find the plate tectonic reconstruction shown in this figure difficult to interpret, because it shows tectonic plates including ridges and continental shelves, rather than a land-sea mask or reconstructed bathymetry. I suggest that the authors use instead a paleogeographic map which would more clearly show the distribution of continents and oceans and the paleodepths of sites; e.g. the Scotese paleogeographic reconstruction maps (Paleomap project); Straume et al. 2020 (paleobathymetry reconstructions available at 1 Ma resolution: https://zenodo.org/record/4193576#.YAb_heB7lXh); or Cai et al 2017 (which includes digital global paleogeographic maps in the supplement, including a 14 Ma reconstruction).

Reply: We will replace the plate tectonic reconstruction in Fig. 1 with the more recent paleogeographic map of Cao et al. (2017), which more clearly shows the distribution of continents and oceans, and provides some (limited) information about the paleodepths of the sites.

Introduction

"The middle Miocene geographic position of Site 747 relative to Antarctica was similar to today"; I found this statement a bit lacking in detail on paleolatitude, setting, etc., so I suggest expanding on this.

Reply: Following the referee's advice, we will add an estimated paleolatitude range for Site 747 from 16 Ma to 12 Ma.

Also the paleodepth of the site is not discussed – could a shallower paleodepth contribute to the relatively warm temperatures you reconstruct compared to modern, and the relatively large changes?

Reply: An effect of a shallower paleodepth on our bottom water temperature record is in principle possible and will be indicated in the main manuscript. However, we would consider such an effect as minor, as we did neither find any evidence for a middle Miocene Site 747 water depth that was shallower than today nor for changes in paleodepth (see also our reply above). In addition, the 2013 World Ocean Atlas dataset (Locarnini et al., 2013) indicates comparably small changes in (annual mean) temperature with depth below ~1000 m (e.g., by around +0.5–1°C from 2500 m to 1500 m water depth) around Site 747. Last, the good agreement of the $\delta^{18}O$, $\delta^{13}C$ and the clumped isotope BWT values from Site 747 with those from Site 761 (Modestou et al., 2020) supports the interpretation that a substantial paleodepth effect on our Site 747 temperature estimates is unlikely.

The $\Delta_{47}$ temperature proxy is well introduced, however given that you list all the potential caveats of the Mg/Ca paleothermometer as applied to benthic foraminifera, I feel the $\Delta_{47}$ proxy gets off quite lightly. A brief summary of the potential impact of diagenesis (dissolution, recrystallization, and overgrowth), burial, or other known non-thermal pro-cesses on $\Delta_{47}$ in benthic foraminifera and their effect on reconstructed temperatures would be useful, even though you discuss this in detail later.

Reply: We fully agree that equal skepticism to all proxies is critical and wish to clarify here that we have attempted to treat all used temperature proxies in an equally critical manner, discussing existing complications and limitations that are relevant for our conclusions thoroughly. However, we will modify the introduction to once more prominently point out the comparably large analytical uncertainties of the clumped isotope thermometer, which in our view pose the main limitation of this technique at the moment. As implied by Referee #1, it is correct that the $\Delta_{47}$ proxy can be susceptible to post-depositional diagenetic processes in certain settings, similar to other more traditional geochemical proxies such as Mg/Ca and $\delta^{18}O$. However, when we investigated diagenetic effects (Leutert et al. 2019), to the best of

our knowledge the only published study specifically investigating the impact of post-depositional diagenesis on foraminiferal $\Delta_{47}$, we found no detectable effects of diagenesis on the $\Delta_{47}$ signatures of middle Eocene benthic foraminifera. This was the case even at pelagic carbonate-rich sites (similar to Site 747) and despite visible signs of diagenetic alteration (e.g., overgrowths of coarse inorganic crystallite), and is supported by evidence based on modeling the effect of diagenesis with reasonable boundary conditions. Compared to the middle Eocene benthic foraminiferal specimens analysed in that diagenesis study, the benthic foraminiferal tests analysed here are from the middle Miocene and thus much younger, making a diagenetic bias even less likely. Of course, there is no absolute certainty when interpreting climate signals from chemical signatures of foraminiferal carbonates as old as the middle Miocene. Therefore, we carefully assessed and documented preservation states of representative specimens (e.g., scanning electron microcopy) and acknowledge this potential source of uncertainty in our proxy discussion. In addition, we will include a sentence on diagenetic effects in the introduction to point this possibility out already earlier, as suggested by Referee #1. However, in the light of the findings from our diagenesis study and the lack of evidence for a diagenetic effect on benthic foraminiferal $\Delta_{47}$ in burial settings comparable to that of Site 747, we think it is reasonable to not put a main focus on diagenetic effects on benthic foraminiferal $\Delta_{47}$ (or Mg/Ca) in the introduction of our study, and instead focus on non-thermal effects on Mg/Ca during biogenic calcite precipitation. These non-thermal effects are clearly much less of a complicating factor for the clumped isotope thermometer in the setting of this study (see also the recent commentary of Evans (2021)).

Methods/Results & Discussion: I think it would be clearer if the Results and Discussion were separated.

Reply: Results and Discussion will be separated.

Age model: I would move the Age Model section up so that it follows the Site Details section.

Reply: Will be done.

In addition, an age-depth plot for Site 747 (in the supplement if necessary) showing all of the different tie points used (magnetostratigraphy, isotope-based, biotratigraphy) and the described hiatus would be very useful.

Reply: We will add an age-depth plot as a supplementary figure summing up the used tie points, in addition to showing sampling, hiatus and core transitions.

Is the assumption that Site 806 sedimentation rates were constant and similar either side of the orbitally-tuned record between 14.1 and 13.3 Ma supported by shipboard magnetostratigraphic and biostratigraphic datums? I would verify this if you have not already, especially given that this is the record that has the most similar trends to your new record. With this assumption, the comparison is not very robust. Presumably the original publication of the Mg/Ca record had age constraints that covered the whole interval?

Reply: We will change our strategy for the Site 806 age model, updating biostratigraphic events from Kroenke et al. (1991) and Chaisson and Leckie (1993) to the GTS2012 timescale (Gradstein et al., 2012) to complement the orbitally tuned Holbourn et al. (2013) age model (instead of assuming constant sedimentation rates from ~16.4 to ~14.1 Ma and from ~13.3 Ma to ~12.3 Ma). We prefer this approach over just taking the original age model of Lear et al. (2015), as the latter age model is not on the GTS2012 timescale and, more importantly, has been optimized for a much longer time interval (~18–0 Ma) and thus not specifically for the middle Miocene sequence at this site. Encouragingly, the updated Mg/Ca-based BWT record from Site 806 shows an even better fit with the clumped isotope BWT record from Site 747. Also, we note that the interpretation of an early cooling across the MMCT (compared to the stepped benthic $\delta^{18}O$ increase) is robust and unaffected by the choice of the Site 806 age model.

The calculation of uncertainties should be briefly described, rather than just referring to the supplement of another paper.

Reply: We will add a brief description of the error propagation in Appendix A.

"Results from adjacent samples are pooled to achieve this number of measurements" Please be more precise about how many adjacent 2-cm samples were pooled together (mean, min, max depth/age intervals over which results were averaged).

Reply: Mean, min and max depths as well as the number of adjacent samples will be added in a supplementary table to make sure that all information requested by the referee is provided (in addition to the supplementary table showing individual replicate measurements with the corresponding depths and ages).

Samples were run on two different machines, but as far as I can see we cannot tell from the figures which data were run on which machine. It might be useful to colour code data points in Figure S3 to show that there are no machine offsets.

Reply: Data points will be coded machine-specific in the supplementary figure. In any case, we note that significant machine offsets are extremely unlikely with our data processing procedure using and normalizing to an identical set of carbonate standards (see also colour-coded data points in Fig. 1 of this response).

[Figure]

*Fig. 1: Comparison of non-averaged Site 747 $\Delta_{47}$ values that are colour-coded for each used mass spectrometer at the University of Bergen (UiB) and ETH Zurich.*

Are the cited external reproducibilities for both of these machines?

Reply: The cited external reproducibilities refer to the performances of all machines that have been used for this study. In the supplementary tables, separate external reproducibilities for each machine and for each standard are listed for all relevant measuring intervals.

As a side note, I feel like Figure S3a should be shown in the main text (maybe as a top panel in Fig. 3), as it shows the raw data upon which all your subsequent data averaging and interpretations are based.

Reply: We are hesitant to move Fig. S3a in the main text, as we do not think that these "raw" $\Delta_{47}$ values can be interpreted in terms of paleoclimate (at least not without further processing/averaging, as pointed out in the main manuscript). Having Fig. S3a in the main text may thus be misleading. We note that this figure will be prominently referenced in the main manuscript.

Fig. 3: horizontal solid lines: averaging intervals; it is not clear to me why the points are not plotted in the middle of the averaging intervals. Is the age of the points weighted towards the highest data density?

Reply: This is correct. As pointed out in the caption of Fig. 3, the position of a plot on the x-axis simply shows the average age over all replicates that were used in a bin. Of course, this implies that for example a sample that has been measured twice (2 replicates) was weighted double. For clarification, we will add this information once more also in the methods (Chapter "2.4 Isotope measurements and data processing").

Why was a 400-kyr moving window approach used rather than a Gaussian-Weighted Filtering approach, as in Modestou et al 2020? I am not sure which method is most appropriate, but the Gaussian-Weighted Filtering approach does seem to smooth out the small-scale features noted by the authors to be caused by scatter in measurements.

Reply: We have tested a lot of different approaches to visually guide the eye including LOESS-based techniques. A LOESS fit has been previously applied to smooth a similar type of clumped isotope record (Leutert et al., 2020). Having weighed up the advantages and

disadvantages of all approaches, we have finally decided for the 400 kyr-moving window approach here, which is comparably simple and easy to understand. This type of smoothing does not only allow for a straightforward comparison between records from different sites and minimizes artefacts caused by uneven sampling, parameter selection and edge effects (LOESS) but also allows for temporally shorter averaging intervals in comparison to other approaches such as Gaussian window filters. Note that Modestou et al. (2020) used a 1000 kyr window size for their Gaussian window filter, which is 2.5 times larger than the 400 kyr window size use here. In our setting, a 1000 kyr window size would make potential biases in the timing of the changes more likely. Using a relatively "simple" moving average without any Gaussian weighing also makes it possible to transparently point to the parts of the record that are based on fewer (<30) measurements and thus less certain (e.g., Fernandez et al., 2017). These advantages of the 400 kyr-moving window approach are weighted more heavily here, than the artefact of minor small-scale features that are not smoothed out. In any case, we note that we provide replicate-level clumped isotope data to allow any reader to reproduce our smoothing or adjust the smoothing for other applications, in addition to applying an alternative approach to visualize the clumped isotope BWT timeseries (binning). Most importantly, our interpretation of the temperature record (early cooling of ~3–5°C, transient smaller warming) appears robust toward different smoothing approaches such as different LOESS fits; we will add a new supplementary figure for illustration.

Add an error bar for Mg/Ca-based temperatures.

Reply: We will add an error bar for Mg/Ca-based temperatures in Fig. 3 illustrating the typical uncertainty introduced by sample reproducibility and calibration errors (±1°C; Lear et al., 2015). In this context, however, it is critical to distinguish between random and systematic errors. Random errors can be relatively easily quantified by comparing multiple measurements. In contrast, the quantitative estimation of systematic errors can be difficult or even impossible with available knowledge, as the cause of the error must be identified and quantified for error estimation. We previously propagated errors and included confidence intervals wherever we considered it possible, meaningful and potentially relevant for interpretation. For Mg/Ca-based temperatures, we had avoided plotting error bars due to known systematic non-thermal influences (such as seawater Mg/Ca or the error in Mg/Ca-based temperature estimates caused by saturation state effects) limiting the informative value of such an error bar. In contrast to Miocene Mg/Ca-based temperature errors, the error in

clumped isotope temperatures is mostly caused by random analytical errors and thus much easier to understand, propagate and quantify.

On Figures 2 and 3, it would be helpful to highlight the middle Miocene climatic optimum and transition intervals, and also the hiatus.

Reply: MMCT, MCO and hiatus will be highlighted in Figs. 2 and 3.

Line 192 – again please specify how large/variable the intervals over which data were averaged are in the text.

Reply: See our previous comment on Page 8 of this reply.

"We note that small-scale features in the moving average curves are likely caused by the scatter in the underlying individual $\Delta_{47}$ measurements, and should not be interpreted as real climate signals" For clarity, please quantify small- scale (<X ∘C) in this sentence.

Reply: "(around 1°C or less)" will be added, as suggested.

Lines 200-203 (and throughout the results and discussion): I suggest citing temperature confidence intervals (± x∘C at x CI) when describing absolute values, this will help to emphasise which trends are significant given the large error bars on $\Delta_{47}$ temperatures (e.g. a 3-5°C cooling is larger than 68% CI).

Reply: The corresponding lines will be adjusted following the advice of Referee #1. At these lines, we will also add confidence intervals for relative changes, whereas in the abstract we prefer to list BWT values without uncertainties, as the exact uncertainty range depends on the exact time interval (whose exact definition is beyond the scope of the abstract). Furthermore, we note that "substantially (~3–9°C) warmer bottom waters" will be changed to "substantially (by up to ~9°C) warmer bottom waters" to be more conservative.

Line 218: How do the recalculated bottom-water temperatures from Site 761 compare to the originally published values?

Reply: The recalculated values are well within uncertainty, and are truly essentially indistinguishable. We will add the values based on the Kele et al. (2015) calibration (updated in Bernasconi et al. (2018), originally used by Modestou et al. (2020)) to a supplementary figure, highlighting the good agreement (well with uncertainty) between the original (Modestou et al., 2020) and the recalculated clumped isotope-based BWT values from Site 761.

Line 229: What artefacts could result from comparing a low-resolution record of discrete samples (each representing maybe 1-2000 years, without knowing if it is a glacial or an interglacial) with a record where each sample integrates hundreds of thousands of years?

Reply: We minimize aliasing in our new clumped isotope temperature record from Site 747, using at least nine adjacent sediment samples and even more separate measurements for each clumped isotope temperature estimate. The large number of foraminiferal tests used for each temperature thus largely prevents aliasing in our Site 747 clumped isotope temperature record. In addition, we calculated clumped isotope temperature using two independent averaging approaches (described in Material and Methods), making our observations for this site even more robust. However, we cannot exclude some degree of aliasing in the Site 806 Mg/Ca-based temperature record (Lear et al., 2015), due to a much lower sampling density and much smaller numbers of foraminiferal tests per temperature estimate (limited temporal resolution of Site 806 Mg/Ca record is cautioned in the Discussion).

Line 269: do the authors have any suggestions as to how to investigate this?

Reply: The specific effect of dissolution on benthic foraminiferal $\Delta_{47}$ could be assessed by laboratory experiments or by analysing samples from a depth transect including sites at different distances from the carbonate compensation depth, similar as has been done in the equatorial Pacific for benthic $\delta^{18}O$ (Edgar et al., 2013).

Line 288: include d18Obw errors in the text. "For the later MCO (15.6–13.9 Ma), our estimates of $\delta^{18}Obw$ range from around -0.3 ‰ to 0.7 ‰´' This statement doesn't really adequately describe the large step changes in reconstructed bottom water δ18Obw at ~14.5 Ma and 13.7 Ma.

Reply: In the previously submitted version, we intended to begin our discussion on bottom water $\delta^{18}$O with a broad overview of the observations from both Sites 747 and 761, and then focus on a more detailed discussion of Site 747 bottom water $\delta^{18}$O and its evolution in the following sentences and paragraphs. We will restructure our discussion of bottom water $\delta^{18}$O including a more detailed description of its temporal evolution and specifically the stepped changes in reconstructed $\delta^{18}$O$_{bw}$ at ~14.5–13.7 Ma (including more details on possible effects on bottom water $\delta^{18}$O, water masses and mechanisms). However, given the possibility of additional biases on $\delta^{18}$O (such as pH or other physiological effects in foraminifera), we prefer to discuss only three approximate $\delta^{18}$O$_{bw}$ ranges without $\delta^{18}$O$_{bw}$ errors bars in the text (with detailed error bars given in Fig 5e).

Line 294: due to their temporal resolution and also due to averaging of many samples probably mixing glacial and interglacial climate states.

Reply: We agree that this addition may make the sentence easier to understand, and will modify the corresponding lines following the suggestion of Referee #1.

Line 326: what was the interpretation of this change in vertical gradient?

Reply: Majewski and Bohaty (2010) interpret this change in vertical $\delta^{18}$O gradient as reflecting a significant decrease in surface water salinity (freshening) across the stepped main increase in benthic $\delta^{18}$O during the MMCT. This interpretation is also supported by our study and previous studies (e.g., Leutert et al., 2020). For clarification, we will include the interpretation of Majewski and Bohaty (2010) more prominently and closer to the text passage, where we are referring to the change in vertical gradient observed by these authors.

---

## Author Comment (AC2) · 24 May 2021

Review "Southern Ocean bottom water cooling and ice sheet expansion during the middle Miocene climate transition" by Leutert et al.

**Response to Referee #2**

Please find below the referee's comments in blue font and the authors' response in black font.

Leutert et al., present the first high-latitude independent bottom water temperature records for the mid Miocene, spanning the bulk of the Miocene climatic optimum and the Miocene climate transition. By using clumped isotope thermometry, the authors circumvent known issues that affect more traditional BWT proxies, such as Mg/Ca.

The main contributions of this manuscript are twofold: 1) By providing independent mid Miocene BWT records, they can evaluate the reliability of deep sea benthic foraminiferal D47 and Mg/Ca records as a BWT proxy. This comparison confirms that D47 is likely an independent temperature proxy that predominantly records BWT, whereas Mg/Ca is affected by non-thermal effects. 2) They show that the main trends in mid Miocene BWT, as reconstructed by D47, are observed at both high and low latitude sites, but are somewhat decoupled from the main trends in ice growth across the mmct. They speculate that regional freshening in the upper water column may be a mechanism to explain this decoupling.

My main concerns are to do with the organization of the manuscript (I have made more specific comments about this below), which can be easily address: - The authors have spread methodological information across the methods, results/discussion and appendix. I found this confusing and will make it hard for readers to later find their methodological approaches. - I found it especially distracting to have Section 3.1 and 3.2 jump between methods, results and discussion. If I were coming back to this paper to find either methodological information or reread the scientific discussion, I would find this frustrating. I think the manuscript would be clearer if the authors could reorganize and group this information better. - The authors also discuss supplementary figures in quite a lot of detail in the main text, so I don't understand why some of those figures aren't incorporated. I am fine with supplementary figures, if the main discussion of those figures is also in the supplement. I found it frustrating to have to go back and forth between the main text & supplement where SI figures were being discussed in detail in the main text. In some cases, the supplementary figures are also only slightly more

expanded versions of the main figures, so I don't understand why the supplementary version isn't used instead of the current main version.

Reply: We thank Referee #2 for the detailed and helpful feedback. We will make a number of adjustments including the reorganization of the manuscript and the inclusion of data from the supplement into the main manuscript. More details can be found below.

Structure concerns aside, the manuscript presents one of the few high-latitude deep sea temperature records of the mid Miocene. I think the manuscript represents a good contribution to Climate of the Past, with interesting implications for both Mg/Ca thermometry and mid Miocene climate reconstructions.

Suggestions in order of appearance:
- Page 1 -
Ln 16: Could you specify which other regions/sites you compare to?

Reply: We will add "from different latitudes" here in the abstract. Although we compile Mg/Ca-based BWT records from a number of sites, we mainly compare our new record to the infaunal benthic foraminifer-based records from Sites 806 and 761 (Lear et al., 2010; 2015) and, of course, the Mg/Ca-based record from the site used in our study, Site 747 (Billups and Schrag, 2002). Therefore, we would consider it potentially confusing for the reader to either list the main sites we used (Sites 747, 806 and 761) or all sites Mg/Ca-based BWT data were compiled from (Sites 747, 806, 761 and 1171) so prominently in the abstract.

Ln 28: In my experience the mmct is defined as the specific benthic isotope excursion ~13.9 Ma (eg Holbourn et al., 2005), much like you'd recognize CM6 or CM5a/b. Could the authors provide a reference for where their definition comes from?

Reply: We agree with Referee #2 that the MMCT is centered at the stepped increase in benthic $\delta^{18}O$ around 13.9–13.8 Ma (e.g., Kochhann et al., 2016), but the onset of this transitional phase was likely earlier and the end later. We will add a reference here (Super et al., 2018) that defines the MMCT from roughly 14.5 Ma to 13 Ma corresponding to our rounded values. We note, however, that there are global compilations suggesting that the

MMCT can be recorded to some extent differently in isotope records (e.g., depending on latitude) (e.g., Mudelsee et al., 2014).

Ln 32: Can the authors include the original publications that produced CO2 records across the mmct, for instance Foster et al., 2012 EPSL?

Reply: Will be done.

Lns 39-41: I don't think the Fairbanks equation is appropriate here, considering that the isotopic composition of the Miocene ice itself was likely different. Can values reported in more recent modelling studies (specifically Gasson et al., 2016, PNAS) possibly provide a better estimate of this?

Reply: Similar to Lear et al. (2015, Pages 12–13), we intend to state with this sentence that assuming a simple linear $\delta^{18}$O-sea level relationship, as proposed by Fairbanks and Matthews (1978), would imply a drop in sea level of roughly 30–110 m. This simplified but also very illustrative example, which is based on transparent assumptions, is followed by a sentence stating a narrower and likely more realistic range for the sea level drop (~20–40 m). This latter (backstripping- and) model-based range is in excellent agreement with the estimate of Gasson et al. (2016, PNAS), who estimated a middle Miocene sea level variability of 30–36 m for a range of atmospheric $CO_2$ between 280 and 500 ppm in combination with a changing astronomical configuration. We will thus include Gasson et al. (2016) as an additional reference here. We will also clarify that the sea level drop estimate of ~20–40 m is based on more advanced methods (than the preceding estimate of ~30–110 m).

Lns 74-75: Can you introduce here what you mean by low data density? Are you hoping to track orbital-scale (eccentricity, obliquity, precession) variability over this time period? This may become apparent later on, but it would be good to introduce this more clearly here, as low vs high data density can mean very different things to different people.

Reply: With low data density we mean here low temporal resolution and potential hiatuses in the middle Miocene record from Site 761 (that is only based on one hole). We agree that our

formulation was not precise enough, and will thus adjust this sentence. We are not aiming for orbital-scale resolution with the clumped isotope data (see also response below), but we prefer to not provide a quantitative estimate of the temporal resolution of Site 761, as we find it difficult to quantify this resolution due to potential hiatuses at the core breaks.

Ln 77: "calcites" should be "calcite".

Reply: "calcites" will be replaced by "calcite", as suggested.

Ln 78 / Ln 83: Please specify that this is in the Indian sector of the Southern Ocean.

Reply: We will add "Indian Ocean sector of" to specify that it is in the Indian sector of the Southern Ocean, as requested.

Ln 88: It would be useful to incorporate the target sample resolution earlier on, especially if you can link the temporal sampling resolution to your goal (still at this stage not clear whether you aim to just get a grasp of long-term changes, or also want to pick up orbital scale variability).

Reply: Following the advice of Referee #2, we will add a sentence in the Introduction to clarify earlier on that our study does not aim at reconstructing orbital-scale variability in BWT.

Lns 88-90: Can you clarify this statement? Do you mean you are using a composite depth scale? Based on the supplementary tables, the authors use a revised mbsf (rmbsf), but they don't actually define that anywhere. It's great to see the authors include the full sample ID and depth, but if they could additionally include the original mbsf as a column in Table S5, that would be better, especially as they seem to refer to the mbsf not rmbsf depths in Section 2.1.

Reply: We will add the original mbsf as a column in this supplementary tables, and specify more precisely where the core expansion at Site 747 is given and has been taken from. We did not use a composite scale for our Site 747 record that is only based on one hole (Hole 747A as specified in Material and Methods).

Figure 1: - It took me a while to understand this figure, and especially to understand that the two maps are of the same area. I was confused as 747 is only highlighted on the left map & 761 only on the right map. Could you potentially adapt this figure to show the full map? Or annotate more clearly on the figure that 1700 m / 2200 m depth are the modern 747/761 water depths respectively? - Also, could the authors add 1171, and ideally U1335/7/8 on the map? I understand the eastern equatorial Pacific sites are hard to fit on with the globe shown as is, but 1171 can definitely be added.

Reply: We will revise Fig. 1 making the titles of Panels a and b more informative (annotating on the figure that 1700 m/2200 m depths are the modern 747/761 water depths, respectively) to more clearly point out that Panel a illustrates the water temperature at the approximate modern water depth of Site 747 and Panel b the water temperature at the approximate modern water depth of Site 761. Amongst other things, this figure aims at providing an overview of water temperature at the depths of the two main sites compared here (Sites 747 and 761 where both Mg/Ca- and absolute clumped isotope-based BWT estimates do exist (this study; Lear et al., 2010; Modestou et al., 2020)). We will also add Sites 806 and 1171 on the inset map with the paleogeographic reconstruction for 14 Ma. However, we prefer to not include Sites U1335/1337/1338 here, as they were only used for the age model and not for our paleoceanographic interpretation.

Lns 106-107: There is some evidence that these species can have different d13C signatures, although this isn't often seen. Has there been any research on clumped isotopes being comparable between the two species? Did you measure the species separately in any of your samples to check they are comparable? Later comment: much of this is later included in the results/discussion section. I found that very confusing and would recommend the authors address interspecies offsets in d18O/d13C/D47 in the methods.

Reply: We measured both *C. mundulus* and *C. wuellerstorfi* separately in 36 sediment samples to assess inter-species $\delta^{18}O$ and $\delta^{13}C$ offsets. We find that in the context of our limited knowledge on benthic foraminiferal $\delta^{13}C$ (and $\delta^{18}O$) offsets in the middle Miocene, the clear offset in $\delta^{13}C$ represents an interesting finding, although not at the very heart of the

our paleotemperature study. We thus include this finding in the results rather than in the methods.

In terms of foraminiferal $\Delta_{47}$, a number of studies using different approaches did not detect species-specific vital effects across a range of species (e.g., Leutert et al., 2019; Peral et al., 2018; Piasecki et al., 2019; Tripati et al., 2015; Watkins and Hunt, 2015). Specifically, the studies of Piasecki et al. (2019) and Modestou et al. (2020) included *C. mundulus* and/or *C. wuellerstorfi* in their assessment. No vital effects on $\Delta_{47}$ are observed even between species with very different stable isotope compositions, i.e. infaunal and epifaunal species. The lack of observed species-specific offsets in $\Delta_{47}$ makes biases caused by pooling measurements from different species unlikely. With this background, we feel confident that we can assume negligible vital effects on benthic foraminiferal $\Delta_{47}$. Another line of evidence supporting negligible $\Delta_{47}$ offsets between the two different species at Site 747 is that individual $\Delta_{47}$ measurements show no discernible offsets between the two species in our data. We will add these information (parallel measurements on both species in 36 samples to assess inter-species $\delta^{18}O$ and $\delta^{13}C$ offsets; no inter-species offsets in benthic foraminiferal $\Delta_{47}$ found in previous studies) including relevant references also in the Chapter "2.3 Sample material" for clarification.

- Page 5 -
Ln 110: Rinsed in DI water?

Reply: Yes, the test fragments were rinsed with deionized water once between each ultrasonication step and at least three times at the end of the cleaning. The sentence will be adjusted accordingly.

Section 2.3 vs Appendix A: As D47 is a key contribution here, I found it confusing that the clumped isotope methods were split between this section and Appendix A.

Reply: Some of the specific details on the methodology and data treatment are only relevant for clumped isotope experts who like to reprocess our data, and would in our opinion be too detailed in the main text, given that our study does not have a technical and methodological focus but rather represents one of the first applications of clumped isotope thermometry to middle Miocene foraminifera. The clumped isotope technique was developed, described in

detail and tested for such applications in earlier studies (e.g., Fernandez et al., 2017; Hu et al., 2014; Leutert et al. 2019; Meckler et al., 2014; Piasecki et al. 2019; Schmid and Bernasconi, 2010). Therefore, we would like to keep the clumped isotope methods relatively concise in the main text, and provide all methodological details needed for reprocessing our dataset in the appendix, supplementary tables and the EarthChem database. For clarification, we will add a sentence in the Section "Isotope measurements and data processing" linking Appendix A more prominently to this section.

- Page 6 -
Lns 143-145: Can the authors provide these recalibrated records in their supplementary data for community use, of course appropriately referencing the original studies? This would greatly help update those records to this more recent calibration. From the supplementary information, it seems the 761 data is included, but not the 1171 data.

Reply: This will be done. Similar to the clumped isotope temperature data from Site 761 (Modestou et al., 2020), we will include the recalculated Site 1171 clumped isotope temperature record (Leutert et al., 2020) using the calibration of Meinicke et al. (2020) as a supplementary table. In this context, we note that although we will update the Site 1171 upper ocean temperature record for consistency; there are no significant differences between these calibrations. Also, we note that the data from our study that is required for recalculation will be made publicly available at the EarthChem archive.

Lns 150-151: GTS2020 was recently published (Raffi et al., 2020), so I would recommend updating the magnetostratigraphic tie points to the most recent timescale. This may not make much difference for the younger interval, but for the oldest reversals used, GTS2020 uses the Chrons recalibrated by Kochhann et al., 2016. As the authors use the Kochhann ages for the d13C based ties, for consistencies sake, I would recommend using the Kochhann et al., 2016/GTS2020 ages for the magnetostratigraphic tie points as well.

Reply: For consistency and comparability, we have tied all records we use to the widely used GTS2012 timescale of Gradstein et al. (2012), similar to Steinthorsdottir et al. (2020) reviewing the state-of-the-start in Miocene climate and ocean circulation. Differences between GTS2012 and GTS2020 (Raffi et al., 2020) were found to be nonexistent to insignificant for the time period of interest in this paper. For example, the Site 1171 age

model of Leutert et al. (2020), which is based on magnetostratigraphic tie points is identical on GTS2012 and GTS2020. In case of Site 747, four of five magnetostratigraphic tie points from ~16 Ma to ~12 Ma (covering the study interval) have identical GTS2012- and GTS2020-ages (C5r/C5An, C5An/C5Ar, C5AA/C5AB, C5AB/C5AC), whereas only one magnetostratigraphic tie point is very slightly (0.020 Myr) younger on the GTS2012 timescale compared to GTS2020 (C5Br/C5Cn). The age of another magnetostratigraphic tie point (C5Cn/C5Cr; age of 16.721 Ma (GTS2012)) that has been used to extend our age model beyond 15.974 Ma for two sediment samples would change slightly more (by 0.084 Myr) when adopting the GTS2020 timescale. Concretely, using GTS2020 ages for the Site 747 magnetostratigraphic tie points instead of GTS2012 would result in the following changes: the ages of 397 (out of 500 measurements) measurements would stay identical, 38 measurements would change by <0.01 Myr and 65 measurements would change by 0.01-0.02 Myr; no age would change more than 0.02 Myr. $\delta^{13}$C-based ties were only considered with a precision of 0.1 Myr. The effect of changing from GTS2012 to GTS2020 would thus be negligible for the timescales of our study and not affecting interpretation and conclusions. However, we would like to point out once more the importance of having all records consistently tied to the same timescale (which is ideally also widely used such as the GTS2012 timescale), wherever possible.

 Ah, the authors discuss interspecies offsets here. I think it would be helpful to discuss in the methods that both species were measured in the same sample to quantify interspecies offsets.

Reply: We will add a sentence in the methods to point out that both species were measured in the same samples in parallel to quantify interspecies offsets in $\delta^{18}$O and $\delta^{13}$C.

- Page 8 -
Figure 2: The authors have done a good job at compiling stable isotope records from different regions; however, I think they are missing some data in the youngest interval. Nathan and Leckie, 2009 (Palaeo3) published low-resolution benthic stable isotope data at 806 and Tian et al., 2017 (Gcubed) and 2018 (EPSL) provides benthic stable isotope data from U1337 between 0 and 16 Ma. Including relevant parts of these datasets could benefit their inter-regional comparison, as they hardly show any data younger than 12.7 Ma in the eastern/western equatorial Pacific Ocean.

Reply: Following the referee's advice, we will include the relevant $\delta^{18}O$ and $\delta^{13}C$ values from the Site U1337 record from Tian et al. (2018) to complement the previous record in the interval younger than 16 Ma (in Fig. 2a and c). We will also include isotope data from Nathan and Leckie (2009). In addition to updating the caption of Fig. 2 with the relevant references, we will generally increase the size of the symbols in Fig. 2 for better readability.

Section 3.2 title: I'm not sure the title really reflects the content of the section. If the authors do reorganize their current section 3 to separate out methods, results and discussion, subsections of this topic could be helpful.

Reply: As requested by Referee #2, we will substantially reorganize Section 3 to, amongst other things, separate methods, results and discussion, and also adjust the corresponding titles (including the title of Section 3.2). Notably, the passage from Line 190 to Line 199 (previously submitted version) will be moved in Material and Methods. Furthermore, we will separate Results and Discussion, as also suggested by Referee #1.

Ln 189: I appreciate the averaged D47 signal is the one the authors want to take forward in their discussion, but can they incorporate Figure S3a into the main manuscript? The D47 data is a key result of the study and they refer to Figure S3 a few times, so it would be better for the reader not to have to flip back and forth between the main text and supplement so often.

Reply: We prefer to have Fig. S3a in the supplement, as, in our opinion, single $\Delta_{47}$ values do not represent the final result and are simply too imprecise to be interpreted in terms of paleoclimate (without averaging). An individual measurement by itself is basically meaningless with our analytical approach (small sample, Kiel device method) and we think that showing the data at individual measurement level in the main manuscript would give the reader a false impression. Note that we will visualize the $\Delta_{47}$ signal using two independent approaches (including fully propagated errors and 68% and 95% confidence intervals). Furthermore, we will generally restructure the Results and Discussion to avoid that the reader needs to consult the supplement to follow the arguments.

First two paragraphs of 3.2 (Lns 190-199 & 200-204): I found the way methods-results-discussion were all mixed together in section 3 to be confusing. Can the authors re- structure it, so the information is easier to find? I found it distracting to continually jump back and forth between methodological information & scientific discussion. The first paragraph is very methodological (190-199), followed by a results paragraph (200- 204), before the authors move on to their discussion. If I were rereading this paper at a later stage for their climatic interpretation and discussion, I would find the inclusion of methodological information in the discussion distracting.

Reply: We agree and will move the passage from Line 190 to Line 199 to Material and Methods. Also, the Results and Discussion will be restructured and separated for clarification (as described above).

- Page 10 -

Figure 3: I appreciate this is not data produced by the authors, but if they could provide error estimates for the Mg/Ca BWT, that would be helpful.

Reply: A typical error for Mg/Ca-based temperatures introduced by sample reproducibility and calibration at Site 806 (Lear et al., 2015) will be added in Fig. 3.

- Page 11 -

Lns 220-222: I don't fully understand what the authors are trying to say here. It currently reads like the authors are saying that Miocene BWT at 747/761 were similar to the present day, but then in the following sentence, they say Miocene BWT were considerably warmer. Do they mean the difference between 747 and 761 is similar to the present day?

Reply: Yes, we referred to the difference between the sites. We propose to rephrase the sentence as follows for clarification and correctness: "Since modern BWTs at Sites 747 and 761 are similar (~1–3°C; see Fig. 1), we expect middle Miocene temperature differences between Sites 747 and 761 to also be small, although the middle Miocene water depths of these sites may have been somewhat different from today."

Lns 229 - 231: Showing these BWT records overlaid would help illustrate this point.

Reply: We are hesitant to show the Site 747 and 806 BWT records overlaid (see Fig. 1 of this reply), due to remaining uncertainties in the absolute ages (especially at Site 806) and also for clarity. Instead, we prefer to focus on the overall pattern of temperature change without stating too strongly that the BWT evolution at Sites 747 and 806 has been the same, also regarding the timing.

[Figure]

**Fig. 1: Overlaid BWT records.** *The curves and values of this figure are identical to those shown in Fig. 3 of the main manuscript.*

Lns 234-245: Can the authors integrate Figure 3 with Figure S4? The authors discussed the various Mg/Ca records included in Figure S4 quite a lot in this paragraph, but only include 2 in Figure 3. I again found it frustrating to have to switch between the main text and supplement to look at data that was discussed in depth in the main text.

Reply: We will move the Site 1171 Mg/Ca BWT curve from the supplementary figures (Fig. S4 in the previously submitted version) to Fig. 3 in the main manuscript to avoid the

need to switch between main text and supplement here, and we will adjust the corresponding figure captions. We prefer to not move the Site 761 Mg/Ca BWTs to the main manuscript as these BWT estimates are in essence represented by the clumped isotope BWT estimates from the same site shown in Fig. 3; a detailed comparison between Mg/Ca- and $\Delta_{47}$-based BWT estimates for Site 761 has already been carried out by Modestou et al. (2020), who found good agreement between the two paleothermometers at that study site. Nevertheless, we intend to still show the comparison between middle Miocene Mg/Ca- and $\Delta_{47}$-based BWTs (Lear et al., 2010; Modestou et al., 2020) in a supplementary figure. In addition, this figure will also illustrate the (very small) effect of using the recent Meinicke et al. (2020) $\Delta_{47}$-temperature equation instead of the recalculated (Bernasconi et al., 2018) Kele et al. (2015) calibration that has originally been applied by Modestou et al. (2020). We further note that we will integrate benthic $\delta^{18}O$ from Sites 747 and 761 into Fig. 3 for better overview and temporal orientation when looking at the BWT curves.

- Page 12 -
Ln 257/Figure 4: I don't understand why Figure S5 isn't used in the main manuscript instead of Figure 4, especially as the inclusion of CO32- data is the main difference between the two figures.

Reply: We did not include Fig. S5 (of the previously submitted version) in the main manuscript, as we see these calculations as a sensitivity test rather than reliable quantitative calculations of bottom water carbonate ion saturation changes, and would like to avoid misunderstandings in this regard. Our goal was to examine the magnitude of saturation change required to bring the $\Delta_{47}$ and Mg/Ca BWT curves together. At present, we cannot fully exclude a dissolution effect on the Mg/Ca and/or the $\Delta_{47}$ signatures of benthic foraminiferal calcite (as pointed out in the main text), largely limiting the practical use of combining benthic foraminiferal $\Delta_{47}$ and Mg/Ca as a way to derive quantitative estimates of carbonate ion saturation changes. Furthermore, we note that the Site 747 Mg/Ca record of Billups and Schrag (2002) has been sampled in low temporal resolution and is based on a comparably small number of foraminiferal tests in the relevant interval, decreasing its representativeness of past environmental conditions (e.g., due to potential aliasing) and also the informative value of the calculated changes in bottom water carbonate ion saturation. In summary, we think that our calculations displayed in Fig. S5 illustrate that changes in bottom water carbonate ion saturation within a reasonable range could actually explain (or at least

contribute to) the observed divergences between Mg/Ca- and $\Delta_{47}$-based BWTs, but do not feel confident enough to put these in the main manuscript.

Lns 257-258: This sentence seemed out of place, as they only discuss 747 in this paragraph and only briefly mention 1171 in the previous paragraph.

Reply: We will remove the sentence.

- Page 14 -
Lns 280-281: This information feels more appropriate for the methods.

Reply: We agree and will move this information to the methods.

Ln 284: "complicating" should be "complicated".

Reply: We will remove "complicating", which is not essential here.

Lns 291-293: What could past differences in the isotopic composition of the Miocene ice sheet mean for these estimates? The authors mention absolute estimates in the introduction, so it seems odd that they do not come back to absolute estimates here after reconstructing d18Osw. If the authors considered the potential past isotopic composition of the ice sheet used in Gasson et al., 2016, they could transfer their new d18Osw estimates to absolute sea level estimates. I think this could be worth including, especially as they mention absolute estimates in the introduction.

Reply: It is true that we mention absolute BWTs in the abstract/introduction but not absolute bottom water $\delta^{18}O$ (and even less absolute sea level) estimates. We only state that systematic biases in bottom water $\delta^{18}O$ estimates may be smaller when using the clumped isotope technique to constrain the temperature component for the calculation of bottom water $\delta^{18}O$ compared to other approaches such as the Mg/Ca technique. In comparison to estimating absolute BWTs, an estimation of a bottom water $\delta^{18}O$ signature that is representative at larger scales (e.g., as a basis to derive global ice volume/sea level changes) is hampered by additional uncertainties including regional imprints of water masses with different salinities, a possible pH effect on benthic foraminiferal $\delta^{18}O$ and additional uncertainties in the equation

linking BWT, benthic foraminiferal $\delta^{18}$O and bottom water $\delta^{18}$O. Estimating past global ice volume/sea level changes would require even more assumptions such as the isotopic composition of the ice sheet, which may have been different from today (as pointed out by the reviewer). Also, we can at this point not be certain to which extent Site 747 located in the Southern Ocean is representative at global scales (or also influenced by Southern Ocean-specific influences). Precise quantitative estimates of the changes in global ice volume/sea level are thus considered premature. We will extend our discussion of the reconstructed bottom water $\delta^{18}$O values adding more information on alternative potential influences such as salinity, pH and the $\delta^{18}$O composition of the ice sheet.

Lns 293-294: This sentence isn't entirely clear; can you rephrase it? Also, could you include the equations used in Figure S6 in the caption or figure itself?

Reply: For clarity, the corresponding sentence will be rephrased. In addition, we will include the equation of Marchitto et al. (2014) in the main manuscript (in Material and Methods) and the equations of Lynch-Stieglitz et al. (1999) and Bemis et al. (1998) in the caption of the supplementary figure, following the referee's advice.

Ln 298: Can the authors rephrase the part of the sentence here to do with orbital parameters? When I first read it, it seems to be undermined by the previous sentence where the authors say that they can't be sure they pick up orbital-scale minima in global ice volume due to the temporal resolution of the D47 records. If they can make it clearer that they mean to look at the influence of longer-term cycles in orbital parameters (e.g., 2.4 Myr/1.2 Myr amplitude modulations), rather than shorter-term orbital cyclicity, that would be helpful.

Reply: We will include the amplitude modulations of the 40 kyr obliquity and the 110 kyr eccentricity (Fig. 5a) to put more focus on the influence of the longer-term orbital cycles that are relevant at our timescales. In addition, we will slightly rephrase the sentence addressing the orbital configuration, but prefer to keep our formulation referring to the observation/interpretation of Holbourn et al. (2005) close to the original wording:

"The main $\delta^{18}$O increase after 13.9 Myr ago occurred during a period when eccentricity declined and amplitude variations in obliquity decreased (Fig. 3). This orbital configuration

would have resulted in an extended period of low seasonal contrast over Antarctica, inhibiting summer melting and favouring ice-sheet expansion." (Holbourn et al., 2005)

Figure 5: - It was hard to read all the details in this figure. Could it be made bigger/wider?

Reply: We will increase the font sizes, optimize the labels and make the figure more compact, but prefer to keep the overall format/width of the figure to be able to fit in one column since we would like to put emphasis on the variability of our reconstructed temperatures (rather than on high-resolution records).

Also, I appreciate the authors are not trying to discuss orbital scale variability, but it could be useful to include one of the higher resolution benthic stratigraphies as a reference for the exact timing of different events. A composite of the records shown in Figure 2 could be used, or the authors could include the new astronomically tuned "zachos curve" from Westerhold et al., 2020.

Reply: We had previously thought about including a higher-resolution benthic stratigraphy as a reference, but prefer to focus on the benthic $\delta^{18}O$ stratigraphies that were measured at the same sites as the available clumped isotope BWT records (Sites 747 and 761). These $\delta^{18}O$ curves are directly comparable to the BWT curves and thus best suitable to assess the (relative) timing of the events on the timescales relevant for this study.

In panel A, could the authors highlight the amplitude modulation of the longer-term eccentricity and obliquity cycles (for instance, as done in Liebrand et al., 2017 PNAS)?

Reply: We will add the 40 kyr-filtered obliquity and the 110 kyr-filtered eccentricity and highlight their respective amplitude modulations. In an additional supplementary figure, we will compare these values to the unfiltered eccentricity and obliquity timeseries.

In the caption, it's unclear whether the authors mean that the "upper ocean temperatures" at 1171 are a shallow bottom water temperature, or a (near)-surface ocean signal, without being aware of the original study.

Reply: Site 1171 upper ocean temperatures were derived from the planktic foraminiferal species *G. bulloides* that is assumed to dwell around 200 m water depth in the Southern Ocean (Vázquez Riveiros et al., 2016). We will add a sentence with this information in the caption of Fig. 5.

- Page 15 -
Lns 319-320: The main d18O decrease certainly occurs after the big drop in temperatures, but I think it also looks like the 747, 761 d18O record is also increasing between the two purple bars, which might indicate that the decoupling is more nuanced than the authors initially suggest.

Reply: We agree with the observation of the referee that $\delta^{18}O$ at Sites 747 and 761 also appears to be broadly increasing between the two purple bars. We will point out in the discussion that the decrease in temperatures indeed coincides with a slow increase in $\delta^{18}O$. However, the temperature decrease derived from $\Delta_{47}$ is much more pronounced, suggesting that there is a counteracting influence on $\delta^{18}O$, likely a decrease in bottom water $\delta^{18}O$. This could be ice retreat (i.e., a decoupling between ice volume and temperature), but also a change in salinity. We will expand the discussion of our results in this regard.

Ln 329: Could you include the location of 1171 on Figure 1.

Reply: Will be done.

Lns 333-335: Could the authors provide a bit more mechanistic detail here about why expansion of the Antarctic ice sheet would result in a freshening of the upper water column? A greater uptake of fresh water in an ice sheet could arguably increase local salinity, not cause freshening. Further explanation might help explain the link the authors suggest.

Reply: Our interpretation of a freshening in certain parts of the Southern Ocean relates to earlier work suggesting a redistribution of fresh water within polar regions in times of increased ice volume and a possible freshening due to increased meltwater input from the growing land-based Antarctic ice sheet and/or sea ice exported equatorward away from the region of sea ice formation (e.g., Adkins et al., 2002; Sigman et al., 2004; Crampton et al., 2016). We will add this information including the relevant references.

---

## Author Comment (AC3) · 24 May 2021

Review "Southern Ocean bottom water cooling and ice sheet expansion during the middle Miocene climate transition" by Leutert et al.

**Response to Referee #3**

Please find below the referee's comments in blue font and the authors' response in black font.

In their study Leutert and colleagues present a record of bottom water changes from ODP Site 747 spanning 16.0-12.2 Ma. The data and the integration of the records to existing geological data reveals fascinating insights into the transition between the Middle and Late Miocene (MMCT). However, in the present manuscript version the authors are retentive in the presentation and discussion of their results. Therefore, it is strongly recommended to further exploit the potential of the study.

Reply: We thank Referee #3 for the constructive comments on our manuscript. We will substantially revise Results and Discussion to be separate and expanded, including more details on all parts of our new bottom water proxy record (e.g., the interval at ∼14.4–13.6 Ma) and on possible underlying mechanisms (e.g., changes in ocean gateway configurations, water mass changes).

Comments:
- Focus of the study is the MMCT. Leutert et al. define this interval as ∼14.5-13.0 Ma, as it contains key changes in the presented records. Given the relatively long definition of this interval it is recommended to define a nomenclature and time intervals for different sub-phases along the key bottom water temperature (BWT) and bottom water d18O (BWd18O) changes. Once introduced they should be used consistently throughout the paper. For instance, statements like 'Our dataset demonstrates that BWTs at Site 747 decreased by ∼3–5°C across the MMCT.' don't do justice to the complexity of the recorded changes, since the full cooling magnitude can be already reached between ∼14.5-14.3 Ma, i.e. even before the phase of major ice growth.

Reply: We will modify the sentence pointed out by Referee #3 to clarify in the abstract that the main cooling preceded the stepped main increase in benthic $\delta^{18}O$ as well as to describe the BWT record from Site 747 in more detail and thus do justice to the complexity of the

reconstructed changes. A similar sentence in the conclusion of the previously submitted manuscript version will also be adjusted to more adequately describe the complex structure of our BWT record.

In Fig. 3, we will highlight the two main sub-phases along the key BWT and bottom water $\delta^{18}O$ changes recorded at Site 747 with coloured bars (1. Phase: bottom water cooling during the early MMCT, 2. Phase: bottom water warming during the later MMCT). In the following Results and Discussion, we will then consistently refer to these two sub-phases. However, we prefer to not go further into detail here and label these phases even more specifically (e.g., for use in future studies), not only due to potential hiatuses (Site 747 record is based on only one drill hole), uneven sampling and age model uncertainties, but also because we do not know the spatial extent of the changes reconstructed at Site 747 (as pointed out in our Discussion and Conclusions).

In addition, we will expand the description of our results. Amongst other things, we will include temperature confidence intervals and link the $\Delta_{47}$-BWT series more directly to benthic $\delta^{18}O$ for temporal orientation (benthic $\delta^{18}O$ values from Site 747 and 761 will be added to Fig. 3). Furthermore, we will generally use the nomenclature for time intervals more consistently, as suggested by the referee. For example, we will replace "middle Miocene greenhouse" in the abstract by "middle Miocene climatic optimum" to use the same term for this period as later on in the text. We will also discuss bottom water $\delta^{18}O$ in a manner that is more in line with the two main MMCT phases seen at Site 747. We further note that we will decrease the x-axis tick mark spacing to 0.2 Myr in the relevant figures, making it easier for the reader to follow the description and discussion of our results.

- The sub-intervals might be chosen to cover the divergence of BWT and BWd18O, including a rapid cooling and abrupt warming into and out-off the phase of minimum BWT (between ~14.3-13.7 Ma). This phase is accompanied by a pronounced decrease (increase) in BWd18O at the beginning (end) of this interval. So far, the focus is towards the end of this interval.

Reply: See previous answer.

- In the current manuscript version, the timing between BWT/ BWd18O changes to upper ocean temperature changes is touched only marginally. However, the timing of these changes can be a key to better differentiate between various forcing mechanisms.

Reply: We will better recapitulate the previously observed coupling of upper ocean temperature and benthic $\delta^{18}O$ at Site 1171 located at high southern latitudes (Leutert et al., 2020), and relate it to our contrasting observation (and interpretation) of decoupled BWT and benthic $\delta^{18}O$ at Site 747 in a substantially revised version of our discussion on regional and global implications.

- Although this paper is a data study, it would be helpful to relate a growing body of relevant model studies to their findings. Interesting aspects might include e.g. the impact of CO2 or ice sheet changes on upper-ocean and BWT changes across the MMCT. Both factors are expected to have different impacts that can support a mechanistic interpretation, since ice sheet changes might have a more heterogenous impact on these temperature records. In this context Section 11.3.5 (Impact of Ice on Miocene Climate) in the recent review of Steinthorsdottir et al. (2020) (doi.org/10.1029/2020PA004037) might be a helpful starting point.

Reply: As pointed out previously, we will substantially expand the discussion of potential mechanisms, relating our observations at Site 747 for example to potential ocean gateway and/or Antarctic ice volume changes. A precise correlation of our new Site 747 proxy record to existing $CO_2$ records is considered to be difficult, as all of these data sets are of rather low resolution and fragmentary. In addition, the $CO_2$ records are partly (in some details) contradictory. Therefore, we prefer to only mention the robust overall decrease in atmospheric $p\mathrm{CO}_2$ across the MMCT (~14.5–13.0 Ma; Foster et al., 2012; Sosdian et al., 2018; Super et al., 2018), but then rather focus on oceanographic changes. For the sake of completeness, we will point out overall decreasing $p\mathrm{CO}_2$ in the interval of this study also in the Discussion.

According the Steinthorsdottir et al. (2020), only two studies (Hamon et al., 2012; Knorr and Lohmann, 2014) have simulated the impact of Miocene ice sheet changes, whereas the bulk of existing paleomodelling studies (e.g., most of those cited in Section 11.3.5 (Impact of Ice on Miocene Climate) of Steinthorsdottir et al. (2020)) focus on older time intervals (e.g., Eocene and Oligocene) characterized by very different boundary conditions (e.g., paleogeography and

ocean gateway configurations, global mean temperature and ice volume) than the middle Miocene. Therefore, in our opinion, these latter modelling studies (e.g., Kennedy et al., 2015; Goldner et al., 2014; Ladant et al., 2014; Kennedy-Asser et al., 2019; 2020) are only of limited use to explain our reconstructed middle Miocene bottom water conditions at Site 747. Furthermore, we note that, although modelling Miocene ice sheet changes, Hamon et al. (2012) focus on testing the impact of varying $p$CO$_2$ and Antarctic albedo on European vegetation during the middle Miocene climatic optimum but not on assessing Southern Ocean mechanisms in detail.

We will therefore include the relevant modelling studies (Hamon et al., 2013; Knorr and Lohmann, 2014) referenced by Steinthorsdottir et al. (2020) in our discussion of the observed changes. The study of Hamon et al. (2013) will be referenced in the context of a potentially closing eastern Tethys gateway inducing circulation changes and thus influencing bottom water conditions at Site 747. We will also add a sentence relating our findings to the interpretation of Knorr and Lohmann (2014), suggesting a complex interplay between winds, ocean circulation and sea ice that may have led to spatially heterogeneous temperature changes in large parts of the Southern Ocean during the MMCT.

In addition, we will relate the results of our study to the recent modelling-based studies of Burls et al. (2021) and Bradshaw et al. (2021). In their synthesis of Miocene climate modelling efforts, Burls et al. (2021) indicate that intermediate to deep waters in the Southern Hemisphere may have been warmer than modern due to differences in ocean currents related to the open Central American Seaway throughout the middle to late Miocene, whereas Bradshaw et al. (2021) point out the importance of the spatial extent of the Antarctic ice sheet affecting the hydrological cycle and deep-water production regions.

---

## Author Response (AR2)

Kiel, 12 September 2021

**Response to Editor and Referee**

Dear Dr. Donnadieu,

We are grateful for a last round of comments on our manuscript. Please find below the referee's comments in blue font and the authors' response in black font.

**Referee #1**

Leutert et al. have thoroughly revised their manuscript, and in doing so have fully taken on board my comments as well as those of two other reviewers. The result is a much clearer manuscript with a fuller discussion of their interesting dataset. Reassessment of the Site 806 age model leads to a more robust comparison with the new record, and Figure 2 clearly shows a good correspondence between the new Site 747 isotope record and higher-resolution, orbitally-tuned records. Figure 1 as well as the paper organisation are much improved. All of my comments have been satisfactorily address, so I am happy to see the manuscript accepted. Small comment on Fig. 3 and 4, I suggest removing the "gradients" from the light green, purple and yellow vertical bars unless they signify something – but this is up to you.

Reply: We have removed the "gradients" from the yellow vertical bars in Fig. 4. However, we would prefer to keep the "gradients" in Fig. 3 to visualize that there is some uncertainty in the exact timing of the onset and termination of the two illustrated time periods.

Fig. 5: The Upper ocean T axis scales for the D47 and TEX86 data need to be the same (9-16°C?) to allow comparison because they are plotted together.

Reply: We have adjusted Fig. 5 to make sure that the upper ocean temperature scales for $\Delta_{47}$ and TEX$_{86}$ are the same (9-16°C). In any case, we note that here the focus is on highlighting mostly relative changes in temperature at Site 1171, similar to Leutert et al. (2020), as absolute temperatures reconstructed from $\Delta_{47}$ and TEX$_{86}$ can be offset to some extent (e.g.,

depending on the temperature calibration for $TEX_{86}$). We further note that we have corrected an error in the measuring unit of Mg/Ca in the caption of Fig. S9 in the supplement ("–0.21 mmol/mol" instead of "–0.21 ‰").

Well done for a very interesting piece of work.

We would like to thank Referee #1 for the encouraging feedback and hope that we have addressed the comments to your satisfaction.

On behalf of all co-authors,

Dr. Thomas Jan Leutert
Corresponding author